# Effect of Heat Treatment on Microstructural Evolution and Microhardness Change of Al-5Zn-0.03In-1Er Alloy

**Jin Cui [1], Zhichao Tang [1], Muzhi Yu [1], Jiajin Hu [1], Xiaoyang Chen [1], Zhengbing Xu [1,2,*] and Jianmin Zeng [2,*]**

[1] Key Laboratory of Nonferrous Materials and New Processing Technology, School of Resources, Environment and Materials, Guangxi University, Nanning 530004, China; 2015301016@st.gxu.edu.cn (J.C.); 2015301066@st.gxu.edu.cn (Z.T.); 1915391048@st.gxu.edu.cn (M.Y.); hu1152627443@163.com (J.H.); xiaoyang_chen@163.com (X.C.)

[2] Center of Ecological Collaborative Innovation for Aluminum Industry in Guangxi, Nanning 530004, China

[*] Correspondence: xuzhb@gxu.edu.cn (Z.X.); zjmg@gxu.edu.cn (J.Z.)

**Abstract:** Adding an appropriate amount of Er element to Al-Zn-In alloys can improve the electrochemical performance of Al alloys; it is convenient to study the electrochemical behavior of the alloy in the rest of our work. However, Er segregation in solid solutions which reduced the comprehensive properties of alloys was difficult to reduce and there was no report on the homogenization of Al-Zn-In alloys. We found that the ultra-high temperature treatment (UHTT) can obviously reduce Er segregation. To explore the better homogenization treatment and the microstructure evolution of Al-5Zn-0.03In-1Er alloy after UHTT, we carried out a series of heat treatments on the alloy and characterized the microstructure of the alloy by optical microscopy (OM), X-ray diffraction (XRD), scanning electron microscopy (SEM), energy spectrum analysis (EDS) and transmission electron microscopy (TEM). The results showed that the main element Er of the Al-Zn-In-Er was largely enriched in grain boundaries after UHTT; the distribution Zn and In was almost unchanged. The as-cast Al-Zn-In-Er alloy consisted mainly of $\alpha$(Al) solid solution and $Al_3Er$ phase. As the temperature of UHTT increased and the treatment time prolonged, the precipitated phase dissolved into the matrix, and there were dispersed $Al_3Er$ particles in the crystal. The proper UHTT for reducing the interdendritic segregation of the alloy was 615 °C × 32 h, which was properly consistent with the results of the evolution of the statistical amount of interdendritic phase, the line scanning analysis and the microhardness. Moreover, the microhardness of the alloy after treatment of 615 °C × 32 h was obviously higher than that of the as-cast alloy because of the anchoring effect of $Al_3Er$ nanoparticles on the movement of dislocations.

**Keywords:** Al-Zn-In-Er alloy; ultra-high temperature treatment; microstructure; line scanning analysis; microhardness

## 1. Introduction

The rapid development of high-tech devices such as aviation and aerospace devices is increasingly demanding on the performance of Al-based alloys. As we all know, different microstructures are exhibited under different treatment, which is extremely significant to control the microstructure evolution during heat treatment in order to obtain excellent mechanical properties. Adding a small amount of rare earth elements can improve the electrochemical performance of the alloys [1,2]. Studies have shown that adding an appropriate amount of Er to the Al alloy can form an $Al_3Er$ phase with the stable $L1_2$ structure [3,4]. According to the Al-Er binary phase diagram [3], there is eutectic $Al-Al_3Er$ on the left side of the phase diagram, and the eutectic point composition is about 6 wt.% Er. Generally speaking, the content of Er in the experimental alloy is much less than 6 wt.%. Zhu, S. et al., [5] found that a small part of the Er solid solution forms a supersaturated solid solution in the matrix and that most of Er is enriched near the interface (beyond the eutectic point

component), which increases the content of Er near the interface. When the alloy reaches a eutectic composition, the eutectic structure of α (Al) and $Al_3Er$ is obtained. Generally, there are three ways that the $Al_3Er$ phase exists in the Al alloy [6]: (1) resolving in the α(Al) matrix in the form of supersaturated solid solution, (2) forming a primary phase $Al_3Er$ or eutectic compound distributed continuously and discontinuously at grain boundaries, (3) precipitating in the form of fine $Al_3Er$ particles during heat treatment, which act as the effective strengthening phase in the alloy. During the solidification and cooling process of the alloy containing Er, one part of the $Al_3Er$ phase satisfies the condition of acting as a heterogeneous nucleation core, which can refine grains. Another $Al_3Er$ segregation phase produces a large number of pinning dislocations and sub-grain boundaries, which will seriously hinder the movement of dislocations, inhibit the nucleation and growth of crystallization, further refine grains and strengthen the alloy [5,7–9]. Al-Zn-In alloys are favored by researchers because of their excellent electrochemical performance and are widely used in the field of corrosion as anode alloys such as seawater [10], deep water [11] and alkaline electrolyte [12]. Li Y. et al., [13] studied the effect of In content on Al-Zn-In-Mg-Ti anodes and confirmed that high In content will lead to uneven dissolution morphology and decrease current efficiency. Therefore, the content of In in Al-based anode alloy is generally less than 0.03 wt.%. Due to the low solid solubility of In in Al-base alloy, the In element mainly exists in the form of a segregation phase. Although the In content is little, it can also play a good activation role on Al-based anode alloy. The anodes containing In without heat treatment can still achieve high current efficiency. Meanwhile, Al-Zn-In alloys have some disadvantages such as uneven surface dissolution, poor mechanical properties, and so on. Our previous work [14,15] found that Er addition plays a role in refining grains in Al-Zn-In alloys. As the addition amount rises, the refining effect gradually decreases and the corrosion resistance of the alloy increases. Another study showed that the current efficiency of Al-Zn-In-*x*Er alloys was increased when the Er content was less than 1 wt.% [16].

However, due to the low diffusion of Er in Al it is difficult to reduce Er segregation in a solid solution and the formation of $Al_3Er$ dispersoids is often inhomogeneously distributed within individual grains [17,18]. Moreover, the cooling rate is so fast during the casting process when the main alloying elements are not well dissolved and unevenly distributed, which is liable to form severe segregation. The nonequilibrium solidified eutectic structure must be homogenized to eliminate or reduce the inhomogeneity of the chemical composition and microstructure [19,20]. Researchers have done much work on the treatment of aluminum alloys. However, research has mainly focused on Al-Zn-Mg [21,22], Al-Mg [23], Al-Cu-Li [24,25], Al-Mg-Mn-Er [26], Al-Zn-Mg-Cu-Zr-Er [27], Al-Cu-Er-Mn-Zr [28] and other alloys [29–32]; the ultra-high temperature treatment of Al-Zn-In alloy after the addition of Er has not been reported.

In previous work, the study of the treatment of Al-Zn-In-Er alloy was attempted. When the alloy was treated at 500 °C, the interdendritic segregation of the alloy showed little change. The segregation phase at the grain boundaries in the alloy microstructure began to decrease until the alloy was heated to above 610 °C. Therefore, to explore the evolution of the phases with increasing temperature and time and its precipitation mechanism UHTT was used in this work to achieve the purpose of homogenizing the alloy. These were basic works for the research group to study the electrochemical behavior of alloys in the future.

Most previous studies have shown that the treatment temperature of homogenization of the aluminum alloys is generally a low or medium temperature. Therefore, the authors were puzzled by the high treatment temperature. It was necessary to study the causes of the high treatment temperature of homogenization. Before the experiment, three hypotheses about the problem were listed: (1) the diffusion activation energy of Er in Al is so low that the diffusion of Er in Al requires a longer time than other elements in Al, (2) there are crystal defects in the alloy, which hinder the homogenization of the alloy structure, (3) the dissolving mechanism of the low-melting-point eutectic phase of the alloy is different from the common aluminum alloy.

There is no doubt that the evolution of the microstructure must have an effect on the properties of the alloy, therefore the microhardness of the alloy was tested to study the hardening effects of $Al_3Er$ phase.

Ultra-temperature treatment and related discussions of Al-Zn-In-Er alloy have not been publicly reported. This work will mainly focus on two parts: (1) to investigate microstructure evolution in the condition of ultra-temperature treatment and try to find the proper treatment process for homogenization, (2) to study the evolution of the $Al_3Er$ phase and how it affects the microhardness of the alloy.

## 2. Experimental Procedure

### 2.1. Material Preparation

According to our previous research [14–16], Al-5Zn-0.03In-$x$Er alloys have good electrochemical properties and it is found that when the content of Er is about 1 wt.% the current efficiency of Al-Zn-In-Er alloy is the best; therefore, we chose Al-5Zn-0.03In-1Er as the experimental alloy. In this work, a master alloy of Al-10 wt.% In was produced by WKDHL-II non-consumable vacuum arc-melting in argon from 99.99 wt.% Al and 99.99 wt.% In. The pure Al and pure In master alloys of Al-10 wt.% Zn and Al-10 wt.% Er were purchased from the Hunan Rare Earth Institute.

The Al-5Zn-0.03In-1Er alloy was smelted by using the above master alloys and pure aluminum in a resistance furnace; 300 g per furnace. The actual composition of the alloy was tested by inductively coupled plasma-atomic emission spectroscopy (ICP-AES) and the results are shown in Table 1. Then, some circular samples were cut from the cast experimental alloy for OM, XRD and SEM tests. Specimens for TEM were cut from bulk specimens, mechanically wet ground to a thickness of about 50 μm, dimpled to 3 μm, and finally twin-jet electropolished in a solution of 25% nitric acid-methanol at temperatures between −30 °C and −20 °C.

**Table 1.** The chemical composition of the experimental alloy (mass fraction, wt.%).

| Elements | Zn | In | Er | Al |
|:---:|:---:|:---:|:---:|:---:|
| The nominal composition | 5 | 0.03 | 1 | Bal. |
| The actual chemical composition | 5.45 | 0.028 | 1.12 | Bal. |

### 2.2. Microstructure Characterization

In the previous study, the reduction of the interdendritic segregation of Al-5Zn-0.03In-1Er alloy under common homogenization conditions was attempted, but it did not work. Therefore, it was decided to homogenize the alloy at an ultra-high temperature. The critical transformation temperature of Al-Zn-In-Er alloy was determined by differential scanning calorimetry (DSC). We chose several temperatures close to the solidus temperature as heat treatment temperatures.

To explore the evolution of microstructure, heat treatments were carried out at different temperatures and treatment times. Then, the evolution of microstructure was investigated by OM, XRD, SEM, EDS and TEM. The distribution of elements in the samples was analyzed by mapping and line scanning analysis.

The microstructure of the alloy was examined by scanning electron microscopy (SEM, Hitachi SU8020, SU8220, S3400N, Hitachi, Tokyo, Japan) and transmission electron microscopy (TEM, JEM-2100, 200 kV, JEOL Ltd., Tokyo, Japan), and the alloy was analyzed by an energy dispersive spectrometer (EDS, Oxford Instruments, Oxford, Britain). Alloy phases were identified by using an X-ray diffraction (XRD, Rigaku D2500 V, Rigaku Co., Tokyo, Japan) operating at 40 kV and 200 mA with Cu Kα radiation. The scanning range was between 20° and 90° (2θ) with a scanning rate of 4°/min.

*2.3. Microhardness Measurements*

The microhardness of the samples after the ultra-high temperature heat treatment was tested to analyze the microstructure evolution of the alloy after UHTT. A microhardness test was carried out with 5 points for each sample by using a Vickers microhardness tester (HWDM-3 hardness tester, Shanghai Taiming Optical Instrument Co., Ltd. Shanghai, China) with a load of 4.9 N and a dwelling time of 15 s.

To reduce the experimental error, five points of tests were carried out in different areas of each sample. The selected areas in the experiment presented a W-type distribution. Figure 1 shows the distribution of test points. We selected areas under a microscope with a magnification of 200× and used the two diagonals of the diamond indentation to calculate the microhardness. The highest and lowest values in each group of data were omitted and the average of the remaining three microhardness values was taken as the microhardness value of the alloy.

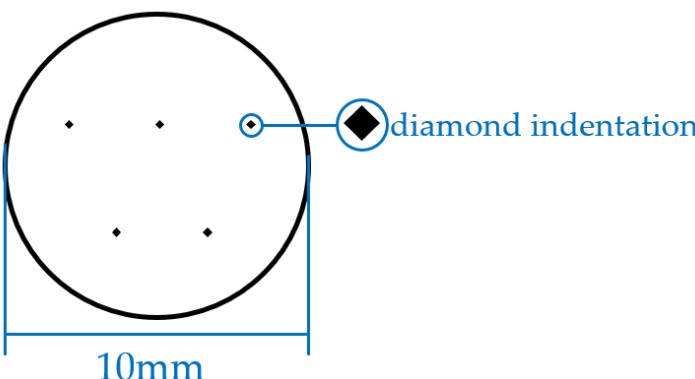

**Figure 1.** Schematic diagram of the distribution of test points.

**3. Results**

*3.1. Characterization of As-Cast Microstructure*

X-ray diffraction (XRD) patterns of the as-cast Al-Zn-In-Er alloy are shown in Figure 2. The XRD peaks show that the main phases of as-cast Al-Zn-In-Er alloy are $\alpha$(Al) and $Al_3Er$; $\alpha$(Al) is an aluminum-zinc solid solution. The actual composition of the alloy in Table 1 showed that there was In element in the alloy. The peaks of the phase containing In is not shown in Figure 2; because, its content is so low and uniformly distributed in the alloy that XRD could not detect In, the segregation of In can be neglected.

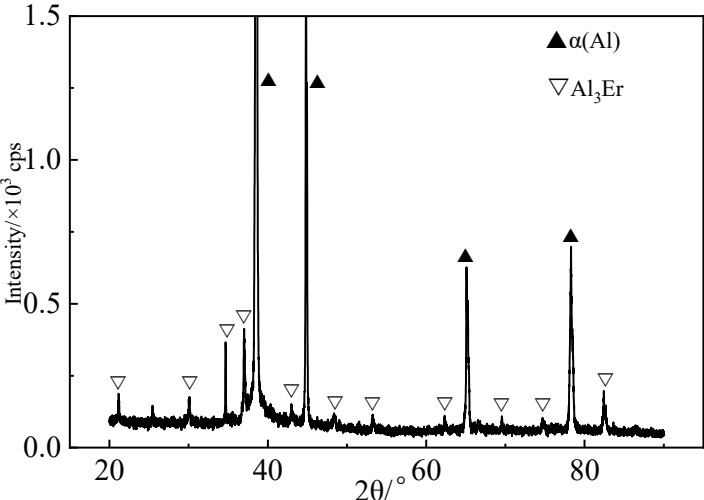

**Figure 2.** XRD pattern of the as-cast Al-5Zn-0.03In-1Er alloy.

The OM and SEM images of the as-cast alloy are shown in Figure 3. As shown in Figure 3a, the as-cast microstructure is composed of an interdendritic α(Al) phase and a eutectic-like phase between the dendrites and the matrix α(Al) is equiaxed. Many coarse continuous nonequilibrium phases are observed in the as-cast alloy. As the white net-like parts show in Figure 3b, there are many massive bright phases distributed at the grain boundaries. According to our previous research [14], it is a typical as-cast microstructure exhibiting interdendritic phase segregation, which requires further heat treatment to reduce the segregation.

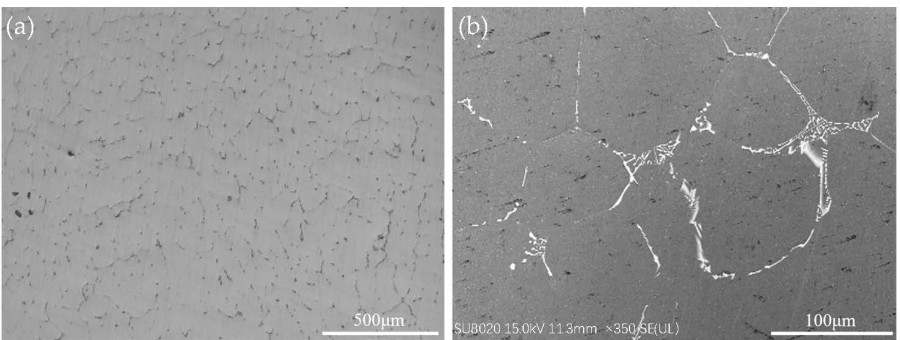

**Figure 3.** Microstructure of the as-cast Al-5Zn-0.03In-1Er alloy: (**a**) OM image, (**b**) SEM image.

The representative microstructure of the as-cast Al-5Zn-0.03In-1Er alloy is shown in Figure 4a, and the corresponding distributions of Al, Zn, In and Er are shown in Figure 4b–e. It is obvious that an Er-rich phase occurred along the Al grain boundaries.

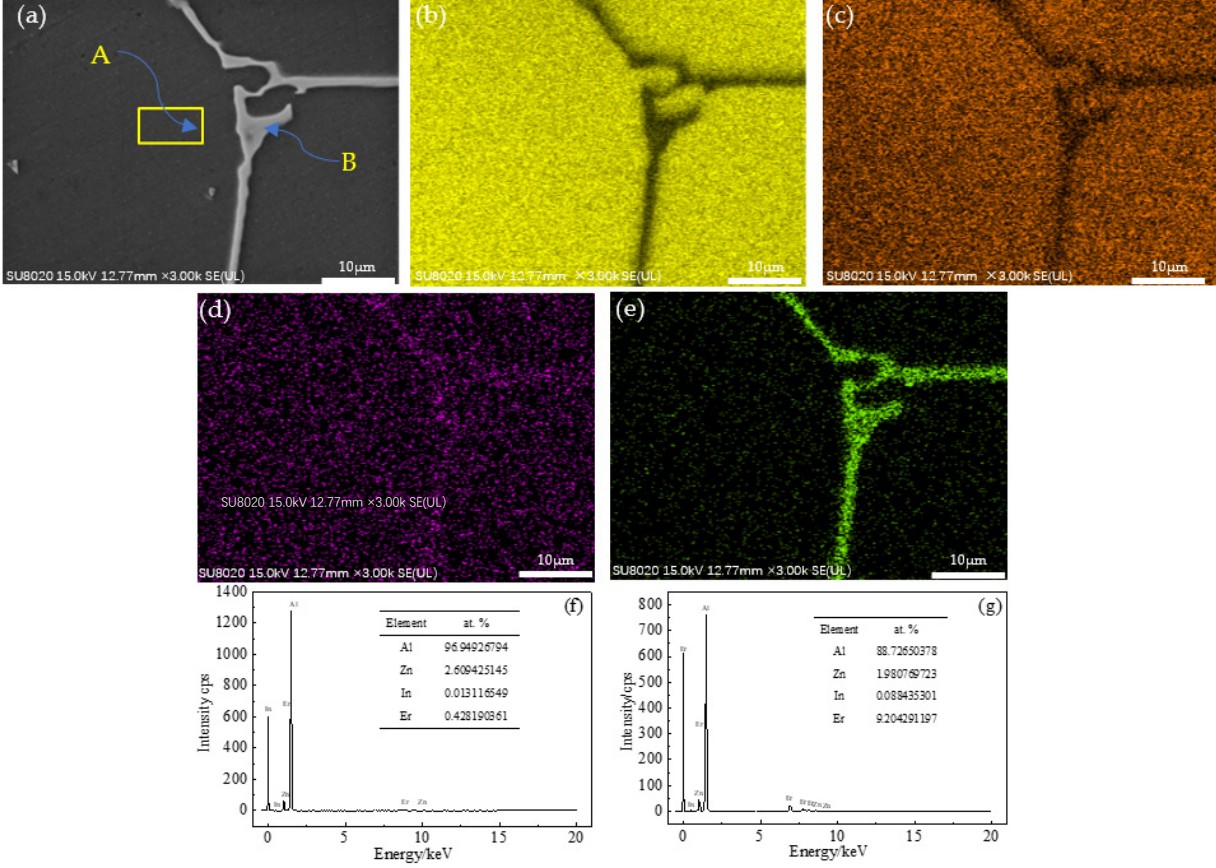

**Figure 4.** Microstructure and the main element distribution in the as-cast Al-Zn-In-Er alloy. (**a**) SEM image, (**b**) Al, (**c**) Zn, (**d**) In, (**e**) Er, (**f**) EDS of Point A, (**g**) EDS of Point B.

Figure 4f,g show the EDS results of the matrix and the second phases in Figure 4a. Combined with the TEM characterization in the section of discussion, EDS analysis reveals that the as-cast alloy is mainly composed of interdendritic α-Al matrix, as marked as point A, and the $Al_3Er$ phase within the α-Al matrix [33], as marked as point B in Figure 4a.

### 3.2. DSC Analysis of As-Cast Alloy

Differential scanning calorimetry (DSC) analysis was conducted with a NETZSCH 404C instrument to determine the UHTT parameters. For DSC analysis, a plate weighing approximately 15 mg from the as-cast ingot was heated to 680 °C under an Ar atmosphere with a heating rate of 10 °C/min. Then, the sample was cooled at the same rate of 10 °C/min under Ar protection. The obtained heating and cooling results are shown in Figure 5.

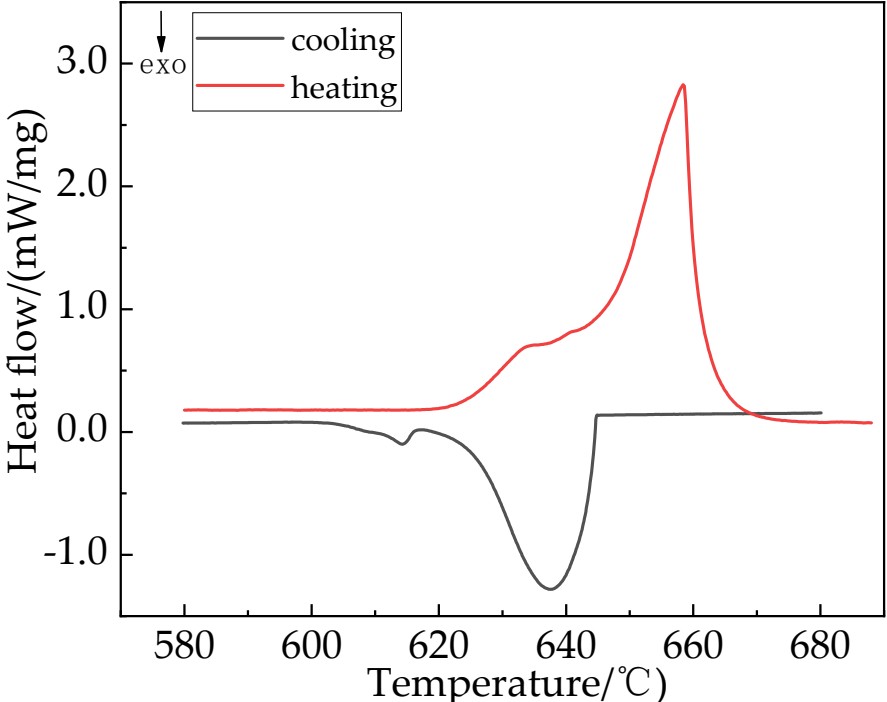

**Figure 5.** DSC curve of the as-cast Al-Zn-In-Er alloy.

According to the Al-Er binary phase diagram [3], the peak at 614.3 °C of the cooling DSC curve possibly corresponds to the melting point of the quasi-eutectic structure (α(Al) + $Al_3Er$) [34], which is similar to general Al-Cu alloy, where the DSC endothermic peak at 532 °C is largely caused by the melting of the $Al_2Cu$ phase [35–37] and the onset at 627.7 °C is related to the initial melting point of the α(Al) matrix [38]. Natively, the UHTT temperature should not exceed 627.7 °C, since the solid solubility of Er in Al is small [14], only approximately 0.6 wt.%, and the diffusion of Er in Al is low [18]. To explore the evolution of the microstructure, the UHTT experiments were carried out at 585 °C, 595 °C, 605 °C, 615 °C and 625 °C for 24 h [4,5,20], followed by annealing. At the UHTT selected temperature, UHTT was performed at 16 h, 24 h, 32 h and 40 h to explore the impact of time on the evolution of the microstructure.

### 3.3. Microstructural Evolution during UHTT

3.3.1. Microstructure Evolution during Different UHTT Temperatures

The microstructure evolution process of Al-5Zn-0.03In-1Er alloy at different UHTT temperatures is shown in Figure 6. Figure 6d–f are enlarged images of the blue box part of Figure 6a–c, respectively.

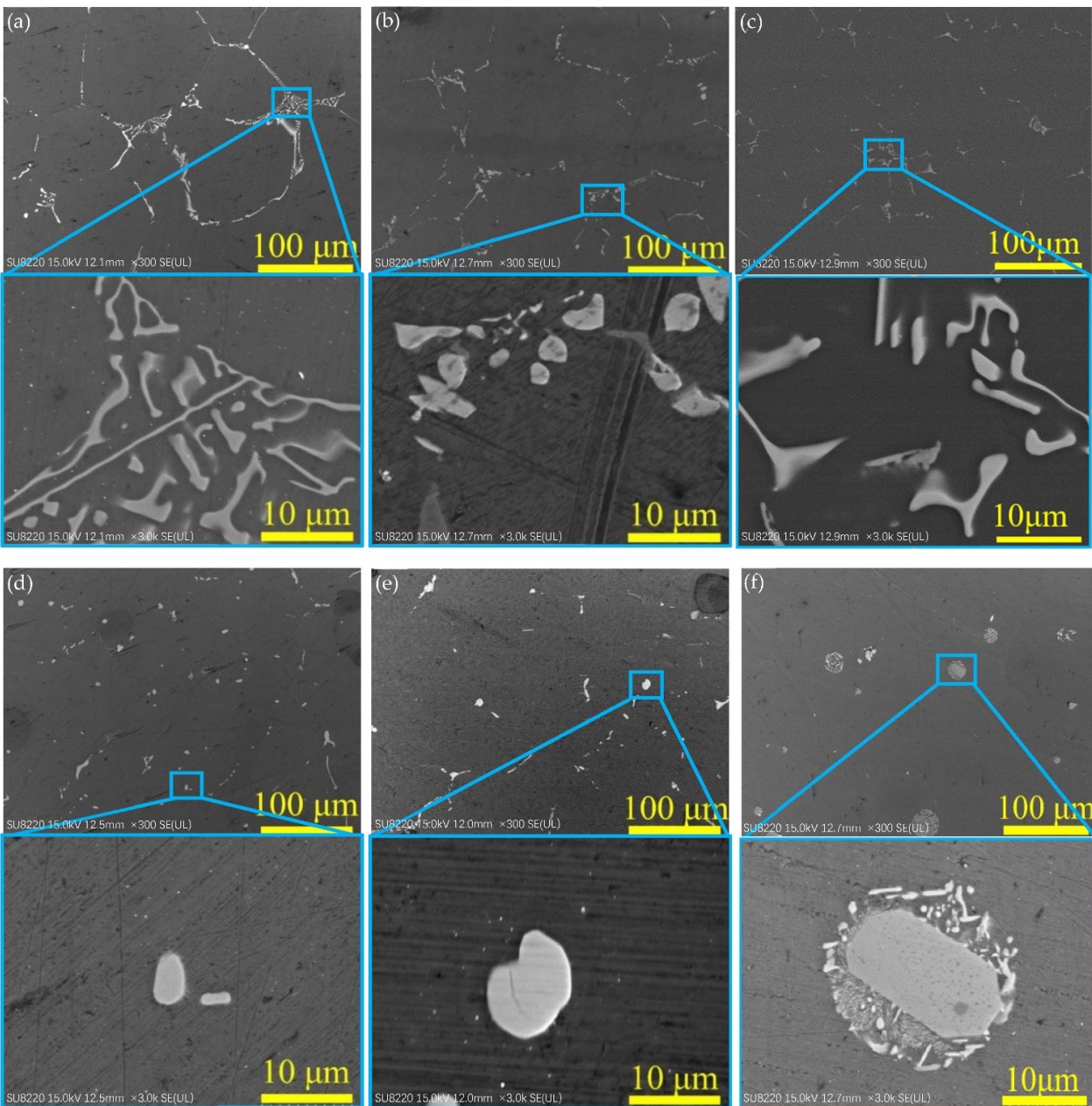

**Figure 6.** Microstructure evolution process of Al-5Zn-0.03In-1Er alloy during different temperature: (**a**) As-cast, (**b**) 585 °C, (**c**) 595 °C, (**d**) 605 °C, (**e**) 615 °C, (**f**) 625 °C.

With the UHTT temperature increasing, in Figure 6 the amount of interdendritic eutectic microstructure in the alloy gradually decreased compared with that of the as-cast state. The net-like interdendritic phase segregation compound was no longer continuous. The precipitate at the grain boundaries became sparse. After UHTT at 605~615 °C for 24 h, the interdendritic network in the alloy became thinner, the nonequilibrium eutectic microstructure and the dendrite segregation were basically reduced (Figure 6c,d) and the microstructure of Al-5Zn-0.03In-1Er alloy was well homogenized.

When the UHTT temperature was further increased to 625 °C, a new space cage-like structure was found in the alloy matrix (Figure 6f). After consulting various studies, it was found that a space cage-like structure has not been reported. The new space cage-like structure is different from the general structure formed by nucleation growth and evolution. Generally, nucleation sites at crystal defects such as grain boundaries, but the new space cage-like structures are located in the interior of grains. Furthermore, it is also different from

the spinodal decomposition structure with non-nucleation phase transformation. For the spinodal decomposition structure, the phase ultimately formed has periodic features that do not fully satisfactorily explain the formation of the space cage-like structure. Therefore, it is necessary to conduct further research in future work. The spherical cage-like structure agrees with the characteristic of an "overburnt structure". To prevent the degradation of alloy properties caused by overburn, we chose 615 °C as the maximum heat treatment temperature instead of 625 °C. Therefore, we excluded the heat treatment at 625 °C.

Combined with the microstructure evolution process in Figure 6, X-ray diffraction (XRD) patterns of the as-cast and UHTT alloys are shown in Figure 7. Each characteristic peak is calibrated with the Miller index. The peaks in Figure 7 show that the as-cast alloys are mainly composed of $\alpha$(Al) and $Al_3Er$. Compared with the XRD patterns of the as-cast alloy, there is no obvious difference in the $\alpha$(Al) peaks in alloy UHTT. After the ingot was ultra-high temperature treated at 585 °C for 24 h, the $Al_3Er$ peaks increased sharply, the supersaturated solid solution decomposed and more $Al_3Er$ phases precipitated in the alloy. When the UHTT temperature exceeded 595 °C, the XRD peak intensity of $Al_3Er$ became weak because of the increase of solid solubility. The content of $Al_3Er$ phases in the alloy began to decrease as the UHTT temperature increased and the $Al_3Er$ precipitated phase began to dissolve back into the $\alpha$(Al) matrix. After UHTT at 615 °C for 24 h, combined with Figure 6e and the $Al_3Er$ peaks at 615 °C, a quantity of $Al_3Er$ phase was dissolved back into the $\alpha$(Al) matrix.

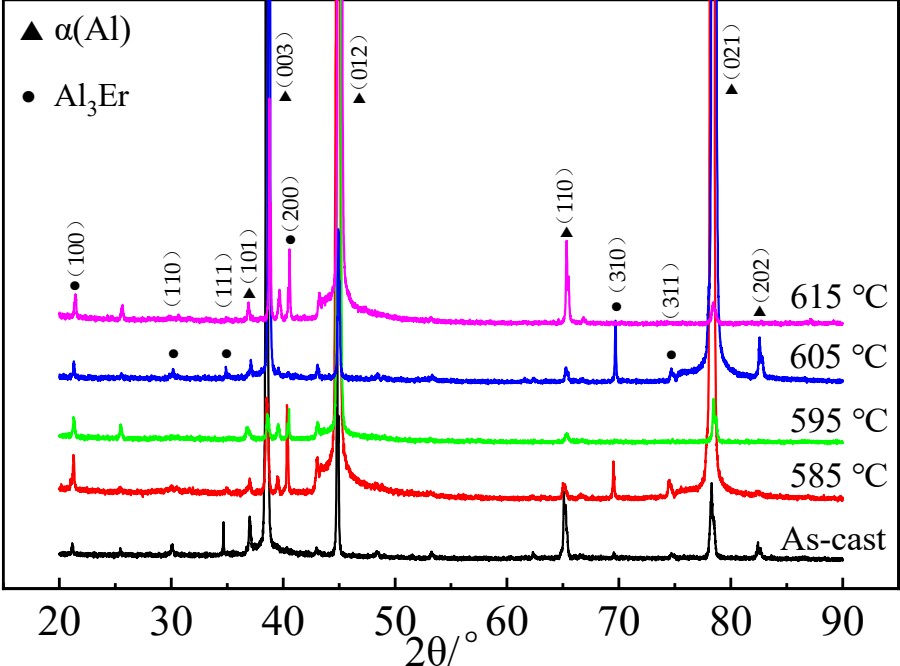

**Figure 7.** XRD patterns of Al-5Zn-0.03In-1Er alloy after treatment at different temperatures for 24 h.

After studying the effect of temperature on the microstructure of the alloy, it was found that 615 °C was worth studying; therefore, a series of studies were carried out to explore the effect of UHTT on the alloy microstructure at different times at 615 °C.

### 3.3.2. Microstructure Evolution during Different UHTT Times

The microstructure evolution process of Al-5Zn-0.03In-1Er alloy during different UHTT times is shown in Figure 8 With the UHTT time increasing. In Figure 8, the amount of nonequilibrium eutectics in the alloy structure gradually decreased from the as-cast state. The reticulated interdendritic phase was no longer continuous. Grain boundaries became sparse. The typical phases of the alloy after UHTT were the discontinuous round particles.

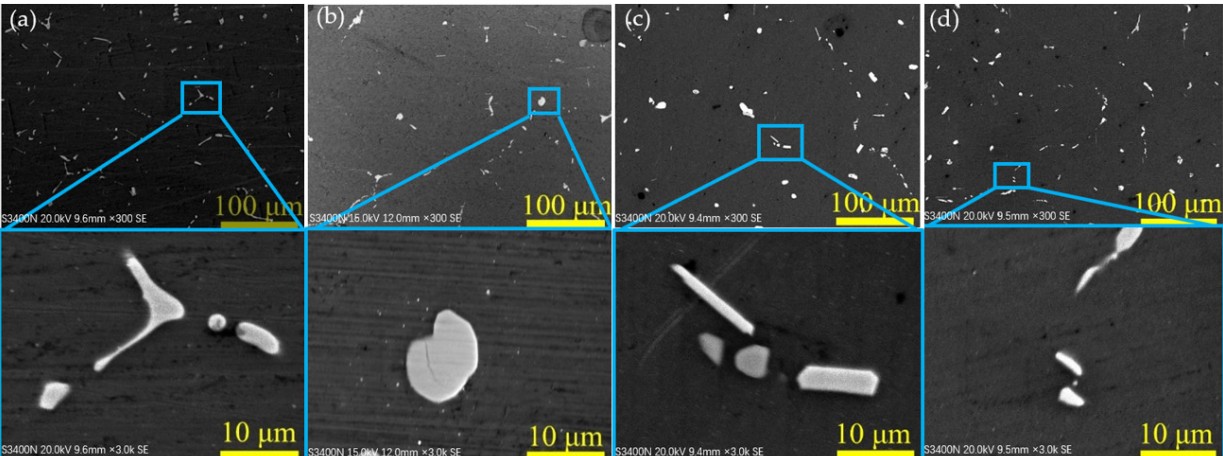

**Figure 8.** Microstructure evolution process of Al-5Zn-0.03In-1Er alloy by treatment at 615 °C for different times: (**a**)16 h, (**b**) 24 h, (**c**) 32 h, (**d**) 40 h.

After UHTT at 615 °C for 32 h, the interdendritic network in the alloy became thinner and the nonequilibrium eutectic structure and the dendrite segregation were basically reduced (Figure 8c). When the UHTT time was further increased to 40 h, the microstructure of the alloy changed little compared with that at 32 h. (Figure 8d). This showed that the second phases of alloys reduced after UHTT at 615 °C for 32 h.

Combined with the microstructure evolution process in Figure 8, as can be seen from Figure 9, the diffraction peaks of α(Al) solid solution and Al$_3$Er phases can be observed in all treated sample at 615 °C for different times. Each characteristic peak is calibrated with the Miller index. Compared with the XRD patterns of the as-cast alloy, there is no obvious difference in the α(Al) peaks in alloys after UHTT. After the alloy was treated at 615 °C for 16 h, most of the Al$_3$Er peaks remained, because the supersaturated solid solution decomposed, and more Al$_3$Er phases precipitated in the alloy. When the UHTT time exceeded 24 h, the content of Al$_3$Er phases in the alloy began to decrease as the UHTT time increased and the precipitated phase dissolved back into the α(Al) matrix, because the solid solubility of Er in α(Al) matrix was not reached. After UHTT at 615 °C for 32 h, the Al$_3$Er peaks nearly disappeared and the Al$_3$Er phases were almost completely dissolved back into the α(Al) matrix or became Al$_3$Er nanoparticles that were difficult to detect. When the UHTT time was 40 h, Al$_3$Er peaks reappeared and Al$_3$Er phases reprecipitated from the matrix, because the re-dissolution of supersaturated solution and the solutes in the matrix were depleted, which reduced the solid solubility of the Al$_3$Er phases.

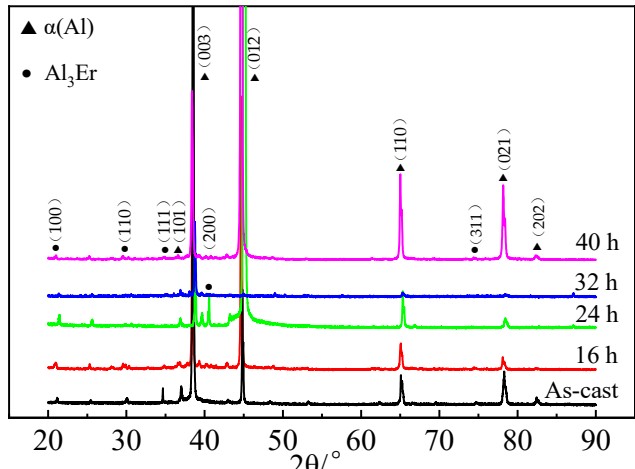

**Figure 9.** XRD patterns of Al-5Zn-0.03In-1Er alloy by treatment at 615 °C for different times.

### 3.4. Line Scanning Analysis

To explore the change of microsegregation in the as-cast and treated alloys, energy-dispersive spectrometer (EDS, OXFORD) linear scanning was adopted and the result is shown in Figure 10. The yellow line in the image expresses the scanning position. The result of the ac-cast alloy is shown in Figure 10a, while Figure 10b–e show the result of the alloy treated at 615 °C for 16 h, 24 h, 32 h and 40 h. The corresponding distribution results of the main alloying elements are shown on the image. Obviously, there is almost no obvious change in the segregation of the main elements between the as-cast alloy and the alloy UHTT at 615 °C for different times. The segregation of In is less evident because of its lower content, and the segregation of In can be neglected. The Zn element distribution is homogeneous from the grain boundary to the inside. However, the concentration of Er content in the segregation phase is significantly higher than in the grain interior and this result indicates the existence of Er segregation. This might be due to the low diffusion coefficient of Er element [18]. Therefore, we can conclude that the diffusion velocity of Zn is faster than that of Er and the segregation of Er exists.

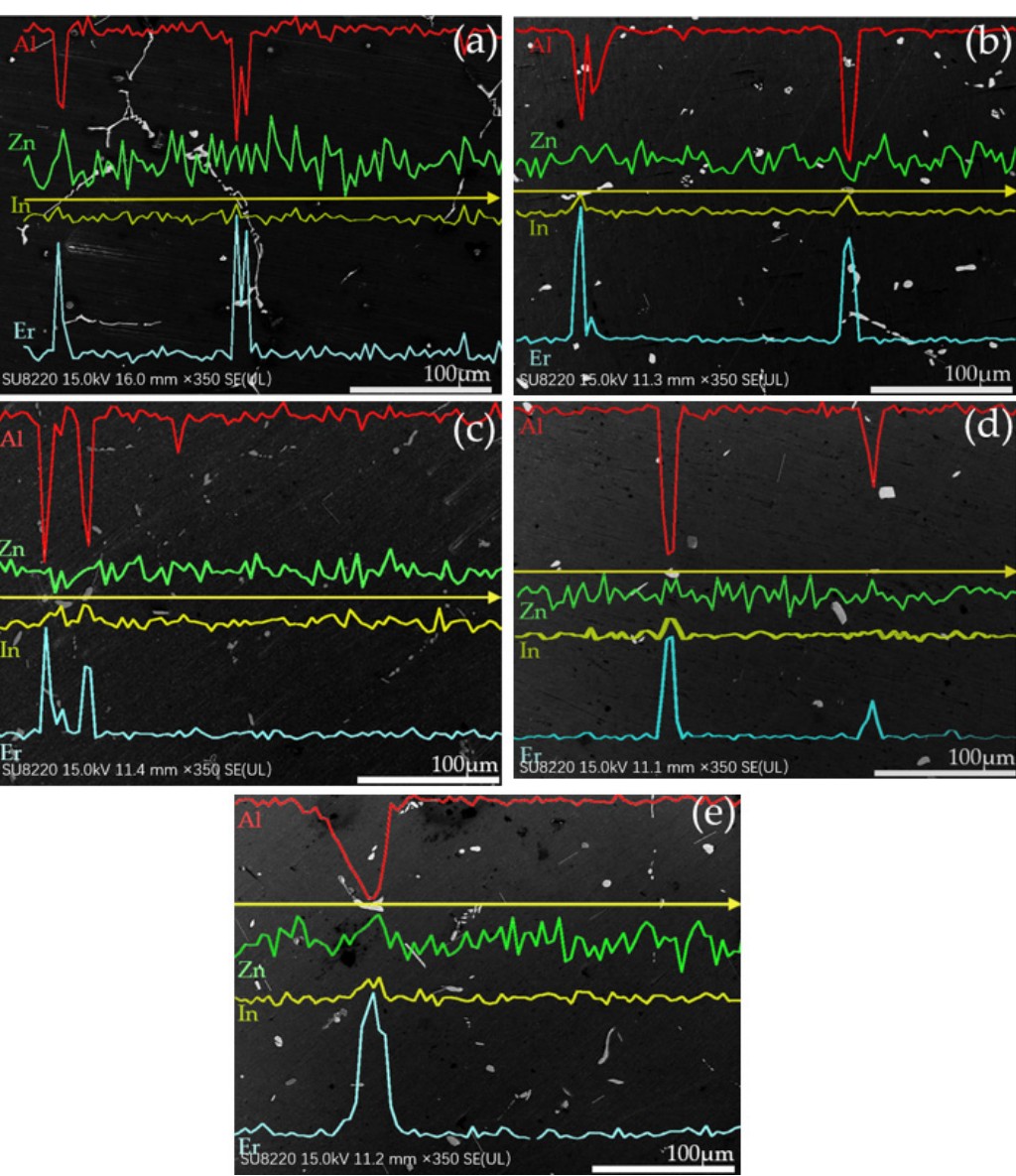

**Figure 10.** Line scanning analysis of the as-cast Al-5Zn-0.03In-1Er alloy and after treatment at 615 °C for different times: (**a**) as-cast, (**b**)16 h, (**c**) 24 h, (**d**) 32, (**e**) 40 h.

### 3.5. Microhardness Analysis

To further confirm the evolution of the second phase (Al₃Er) during different treatment times, the microhardness tests were performed. The average microhardness of Al-5Zn-0.03In-1Er alloy under different heat treatment time is plotted in Figure 11. We added error bars to the line chart. Figure 11 shows that the microhardness first increased to the maximum when the treatment was 32 h and then hardly changed with increasing treatment time. When the UHTT time was 32 h, the microhardness increased by 15.5% more than that of the as-cast alloy.

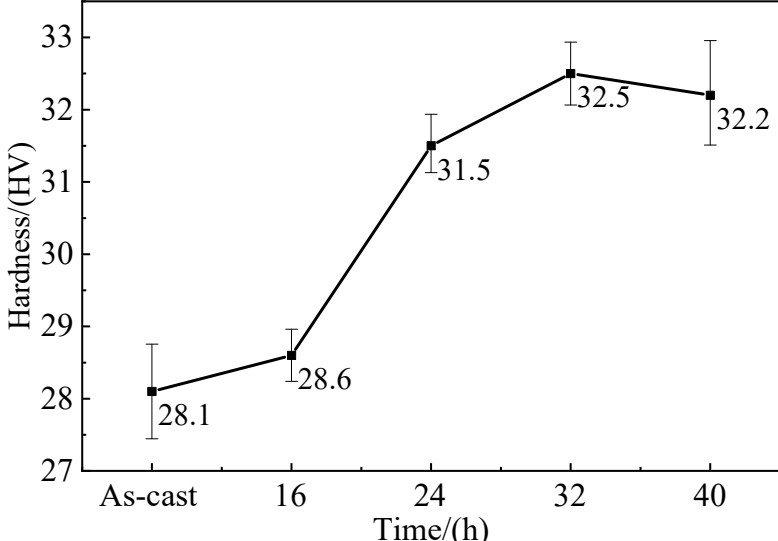

**Figure 11.** Microhardness of the alloy after treatment at 615 °C for different times.

## 4. Discussion

The UHTT usually has three purposes: (1) to reduce dendrite segregation during solidification and to homogenize the distribution of alloy composition, (2) to resolve the supersaturated solid solution formed under casting condition, (3) to reduce the macroscopic stress in as-cast alloys. A good treatment can achieve the above objectives well without over-heating or overburning the alloy microstructure, thereby reducing the overall performance of the alloy. Therefore, the treatment temperature should not be too high to damage the properties of the alloy. However, the segregation phase at the grain boundaries in the Al-Zn-In-Er alloy microstructure began to decrease until the alloy was heated to 615 °C. Therefore, it is crucial to have a clear understanding of the ultra-high treatment temperature.

The main parameters of the treatment system are heating temperature and holding time. It can be known from Fick's first law: in unit time the amount of material diffused through a unit area is proportional to the concentration gradient of material perpendicular to the direction of the cross section [39], that is:

$$J = \frac{dc}{dx} \tag{1}$$

where diffusion flux ($J$) is defined as the mass (or, equivalently, the number of atoms) diffusing through and perpendicular to a unit cross-sectional area of solid per unit of time. $D$ is called the diffusion coefficient, which is expressed in square meters per second. Temperature has a most profound influence on the coefficient. The temperature dependence of diffusion coefficients is related to temperature according to:

$$D = -D_0 \exp\left(-\frac{Q}{RT}\right) \tag{2}$$

where $D_0$ is a temperature-independent pre-exponential (m$^2$/s), $Q$ is the activation energy for diffusion (J/mol), $T$ is the treatment temperature and $R$ is the gas constant (8.314 J·mol$^{-1}$·K$^{-1}$).

Therefore, the higher the heating temperature the faster the diffusion rate. As the diffusion process progresses, $J$ gradually decreases, indicating that the diffusion rate decreases from fast to slow with the extension of holding time. Therefore, the microstructure changes significantly in the early stage of treatment, while the microstructure does not significantly change when the holding time continues to be extended, as shown in Figure 8.

Since the treatment is closely related to diffusion activation energy($Q$) and diffusion constant ($D_0$), many studies have been consulted. Some key data are listed in Table 2. In addition, it was reported that the diffusion coefficient of Cu in Al was much lower than that of Mg and Zn at the same temperature [34,40] and the diffusion coefficient of Er is much lower than that of Cu. Therefore, it is not difficult to find that the $Q$ and $D_0$ of Er in Al are relatively small and natively diffusion requires higher temperatures than other common Al alloys.

**Table 2.** The $Q$ and $D_0$ of some diffusion systems.

| Diffuser in Al | $Q$ (kJ/mol) | $D_0$ (m$^2$/s) | References |
|:---:|:---:|:---:|:---:|
| Mg | 130 | $1.2 \times 10^{-4}$ | [41] |
| Cu | 136.8 | $8.4 \times 10^{-6}$ | [31] |
| Er | $77.2 \pm 5.3$ | $(4.3 \pm 2.2) \times 10^{-12}$ | [42] |

The Al-5Zn-0.03In-1Er alloy treated at 615 °C for 32 h was characterized by TEM to further determine the composition of the precipitated phase in Figure 12. The TEM image of the alloy is shown in Figure 12a and the high-resolution TEM characterization of both sides of the boundary is shown in Figure 12b. Figure 12b shows that the second phase is on the left side of the boundary, the matrix is on the right side; results analyzed by Digital Micrograph (DM) software. After performing the Fourier transform of the image and measuring the distance, the distance between the crystal planes of (100) of the second phase is about 0.421 nm, which accords with the distance between the crystal planes of Al$_3$Er, and the distance between the crystal planes of (012) of the matrix is about 0.199 nm, which satisfies the distance between the crystal planes of an Al-Zn solid solution. The selected area electron diffraction patterns (SAEDP) of the second phase and the matrix were shown in Figure 12c,d. After calibrating the SAEDPs, the result is the same as the crystal face indices of the Al$_3$Er and Al–Zn solid solution.

Figure 13 displays a TEM image of the as-cast Al-5Zn-0.03In-1Er alloy and after treatment at 615 °C for 32 h. According to Figure 13a, there were many dislocations with the similar orientation in the as-cast alloy. Figure 13b shows that there were fewer dislocations and that the second phase particles were diffusely distributed. The selected area electron diffraction patterns (SAEDP) of the particles show that a large number of Al$_3$Er particles with a size of 10~20 nm precipitated out. This is basically consistent with the above analysis of results. The cooling rate of the alloy was fast in the casting solidification process, therefore the Er element in the ingot did not precipitate in sufficient time, which made the ingot in the supersaturated solid solution state. Generally speaking, the supersaturated solid solution was thermodynamically unstable, therefore a large amount of Al$_3$Er phase precipitated in the heat treatment process. With the increase of heat treatment time, most of the Al$_3$Er phase gradually dissolved in the matrix, and the remaining dispersed Al$_3$Er particles were distributed in the grains.

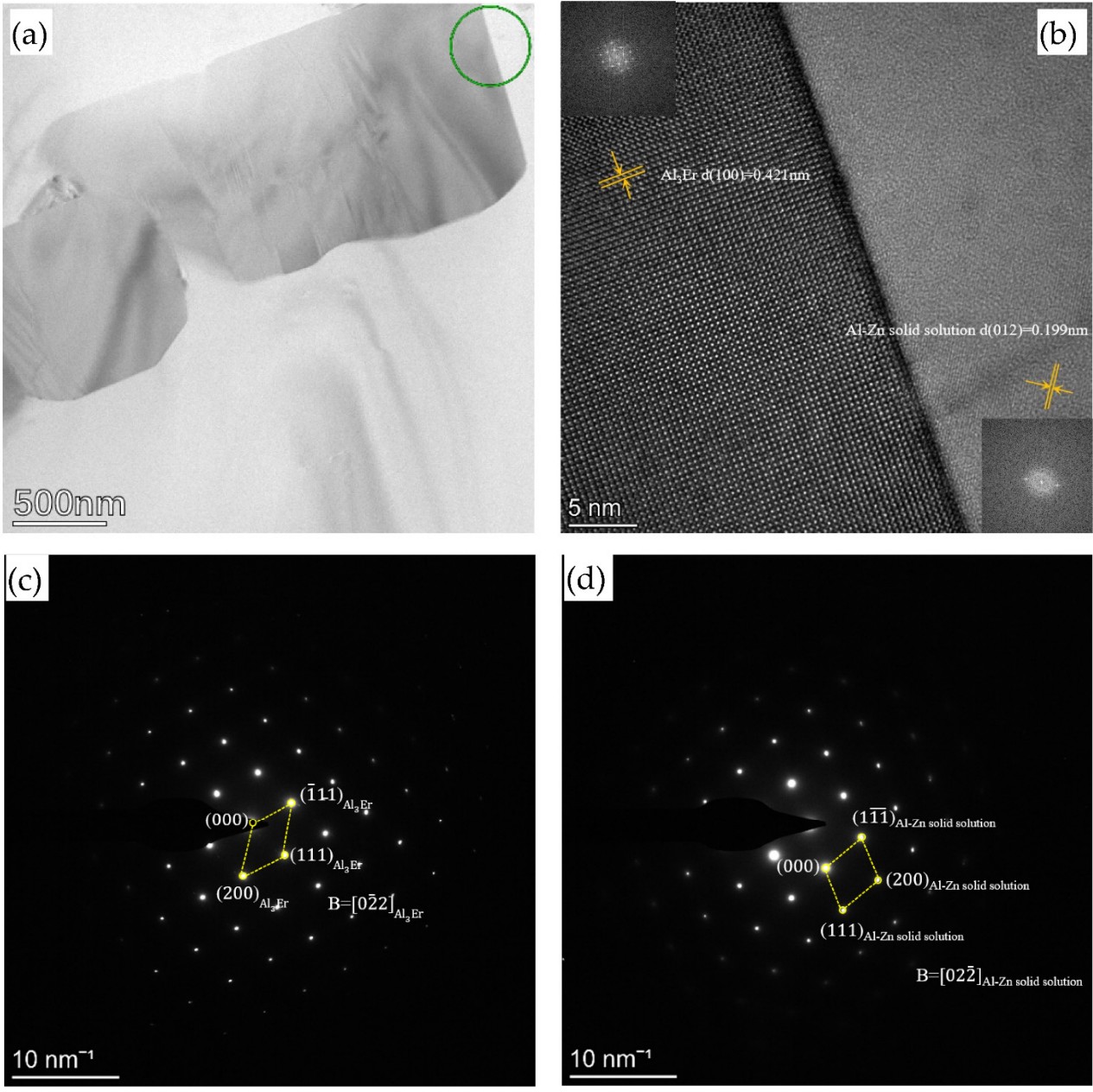

**Figure 12.** TEM images of the Al-5Zn-0.03In-1Er alloy treated at 615 °C for 32 h: (**a**)TEM image, (**b**) HRTEM, (**c**) the SAEDP of the second phase, (**d**) the SAEDP of the alloy matrix.

The $Al_3Er$ phase has an extremely high melting point and strong stability at a high temperature. In addition, there was a phenomenon of dislocation tangle in the alloy (Figure 13b), because $Al_3Er$ particles hinder and pin down the movement of dislocations. $Al_3Er$ particles are pointed by yellow arrows in Figure 13b.The existence of dislocation tangle had an important influence on the diffusion, anti-deformation ability and phase transformation of the alloys. In general, diffusion can proceed faster along dislocations, however the anchoring of dislocations due to precipitation phases ($Al_3Er$) can reduce the diffusion rate and hinder the slip of dislocations. This is a main cause of the high treatment temperature and the higher microhardness of the alloy after UHTT. This is consistent with the trend of the microhardness curve in Figure 11.

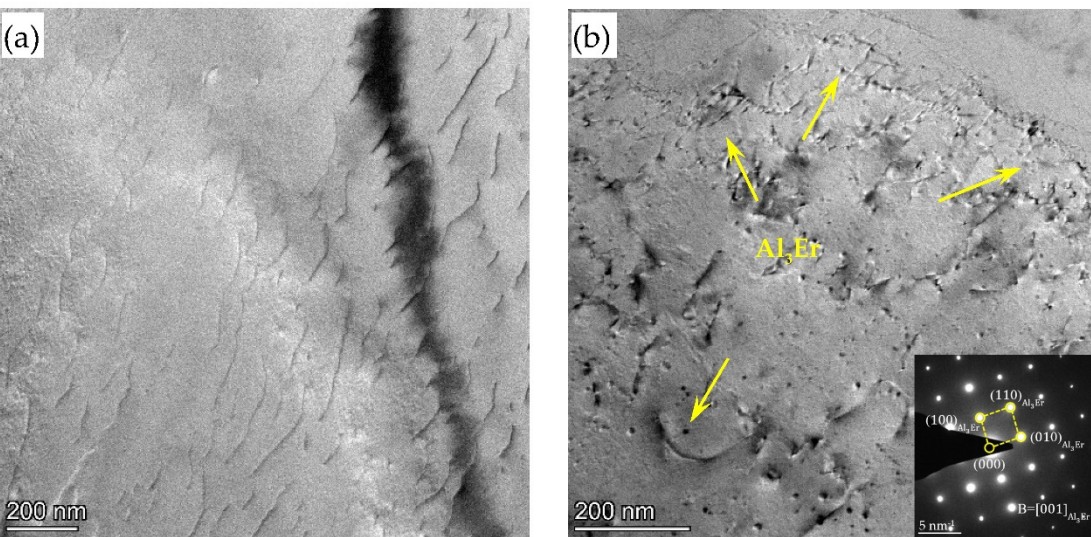

**Figure 13.** TEM images of the Al-5Zn-0.03In-1Er alloy: (**a**) as-cast, (**b**) 615 °C × 32 h.

Since the atomic radius of the rare earth element is large, the solid solubility of the rare earth element in Al is generally small. The atomic radius of Er is 0.1757 nm, which is 23% larger than the atomic radius of Al. According to the Hume-Rothery rule, when the solute and solvent atomic radius differ by more than 15% only a solid solution with a small solid solubility can be formed in the alloy system; therefore, under equilibrium conditions the solid solubility of Er in Al is limited. This is also the reason why the treatment requires an ultra-high temperature.

Figure 14 shows the evolution of the statistical amount of the interdendritic phase after treatment at 615 °C for different times. The statistics of every condition were analyzed by using Image-Pro Plus software and by selecting more than five images at a same magnification. The grain boundary phase first decreased sharply within the initial 16 h and its amount decreased slightly with the further prolonging of treatment time.

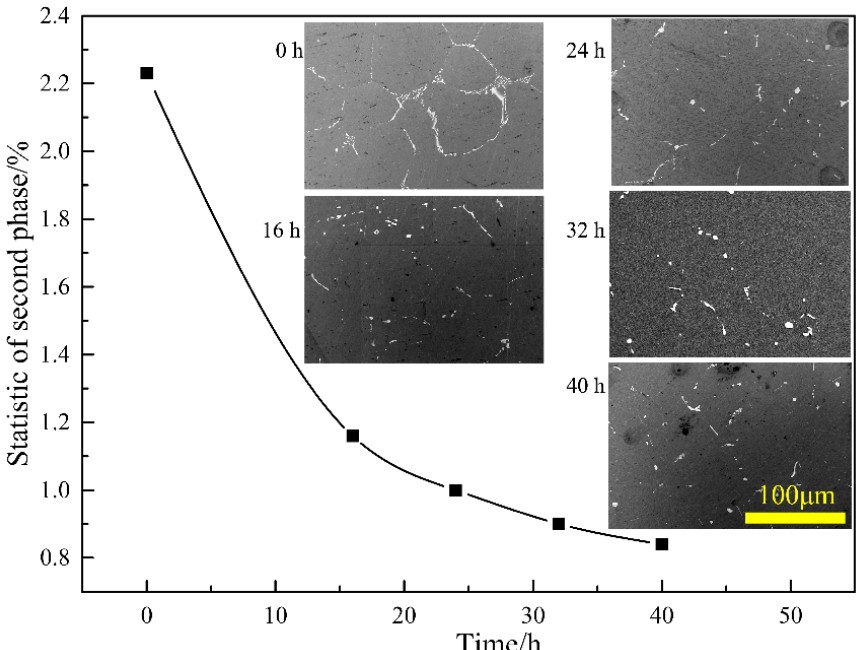

**Figure 14.** Evolution of the statistical amount of interdendritic phase after treatment at 615 °C for different times (0 h, 16 h, 24 h, 32 h, 40 h).

The microhardness of alloys is mainly dependent on three factors: grain size, strengthening effect and second phases. The secondary phases gradually dissolved into the matrix, which resulted in a uniform distribution of residual granular secondary phases. These residual granular secondary phases could also act as a strong obstacle to dislocation motion. Meanwhile, significant grain growth was not observed. Therefore, the strengthening effect was believed to be a dominant factor for the microhardness at the initial stage of treatment.

The addition of alloying elements causes lattice distortion and the precipitation phase has a significant anchoring effect (Figure 14); both can hinder the movement of dislocations, which in turn produces a strengthening effect. Therefore, the microhardness increases significantly at first.

The existence of dislocations also has an important influence on the mechanical properties of the alloy. In the Al-Zn-In-Er alloy, the precipitated phase ($Al_3Er$) is an ordered phase. The dislocations in the ordered alloy are all superdislocations. The plastic deformation of alloys requires the simultaneous movement of two dislocations of superdislocation. A larger external stress is required, which is ordered strengthening. In this case, the dislocations of the alloy must pass through the precipitation phase (cutting the precipitate phase) to make the alloy plastic deform. This process will destroy the ordered arrangement of the precipitated phase, therefore higher external stress is required.

However, when the treatment time was up to 40 h, the influence of solid solution strengthening may be reduced because the matrix was depleted of solutes, which was caused by the precipitation of $Al_3Er$.

On the other hand, in the $Al_3Er$ phase a type of order solid solution with a stable $L1_2$ structure can produce an order-hardening effect. In the previous study, the microhardness first increased and then decreased with an increasing order degree; therefore, the microhardness results are consistent with the law.

## 5. Conclusions

In this study, the microstructural and thermal analysis of as-cast Al-5Zn-0.03In-1Er alloy were carried out. A series of heat treatments were established and the effects of different heat treatments on the microstructure and the microhardness of Al-5Zn-0.03In-1Er alloy were investigated to obtain the better homogenization heat treatment of the alloy. The microstructure evolution and the microhardness change of Al-Zn-In-Er alloys were investigated following ultra-high temperature treatment (UHTT). Explanations for these phenomena are given. The following conclusions could be drawn:

(1) The interdendritic segregation exists in the as-cast Al-Zn-In-Er alloy. With the increase of UHTT time, the main element Er is largely enriched in grain boundaries and its concentration decreases from the grain boundary to the inside, but the changes of Zn, In are not obvious.

(2) The dissolvable precipitated phase in the as-cast Al-5Zn-0.03In-Er alloy contains $\alpha(Al)$ and $Al_3Er$ phases. The melting point of the precipitated phase is lower than that of Al, Er phases. The precipitated phase gradually dissolves into the matrix at 614.3 °C and $Al_3Er$ particles are dispersed in the crystal after treatment.

(3) The proper UHTT process is 615 °C $\times$ 32 h, which is consistent with the results of the evolution of the statistical amount of the grain boundary phase and the line scanning analysis. When the UHTT is 615 °C $\times$ 32 h, the microstructure of Al-5Zn-0.03In-1Er alloy is well homogenized and the microhardness of the alloy is 32.5 HV, which increases by 15.52% more than that of the as-cast alloy. The reason may be the significant anchoring effect on the movement of dislocations, the solid solution effect or the order-hardening effect of the $Al_3Er$ particles.

## 6. Future Research

In addition to the above conclusions, this work has the following prospect for the new space cage-like structure: after a preliminary judgment, the space cage-like structure is not fully consistent with the characteristic intergranular of a eutectic-like structure. Moreover,

the spinodal decomposition mechanism cannot completely explain its transformation mechanism. Therefore, it is necessary to conduct further research that will include the following major aspects: i) the space cage-like structure formation and growth mechanism, ii) the specific ways of nucleation, growth and impingement of this kind of solid-state phase transformation.

**Author Contributions:** Conceptualization, Z.X. and J.C.; software, J.C.; methodology, J.C.; formal analysis, X.C., Z.T. and M.Y.; investigation, J.H. and X.C.; resources, M.Y. and Z.T.; writing—original draft preparation, J.C.; writing—review and editing, J.C.; visualization, J.C.; supervision, Z.X.; project administration, J.H.; funding acquisition, J.Z. and Z.X. All authors have read and agreed to the published version of the manuscript.

**Funding:** This work was supported by the Natural Science Foundation of China (Grant No. 51961008), the Regional Joint Fund of National Natural Science Foundation of China (U20A20276) and Guangxi Natural Science Foundation (2020GXNSFAA297269).

**Data Availability Statement:** Data supporting reported results can be found in this paper.

**Acknowledgments:** The authors are grateful to our former group members Xiaoyang Chen for the differential scanning calorimetry measurement and analysis discussion.

**Conflicts of Interest:** The authors declare no conflict of interest.

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
