# Peer review of "Effect of Heat Treatment on Microstructural Evolution and Microhardness Change of Al-5Zn-0.03In-1Er Alloy"

_metals, doi:10.3390/met12030370_

Round 1

Reviewer 1 Report

The topic of the paper is interesting. But the investigation should be significantly reworked. Microstructure and phase composition was not investigated and adequately described.  The most important points are indicated below.

  1. “Studies have shown that adding an appropriate amount of Er to the Al alloy can form Al3Er phase with the stable L12 structure 38 [3, 4].”

The origin of the Al3Er phase should be indicated.

The Al3Er phase formation in the Al-Er alloy first of all follows from the binary phase diagram.

Al3Er phase formation in the Al alloys was shown significantly earlier.

Please change an approach to site the literature.

  1. The grain refining effect of Er in the Al alloys are investigated by more researchers than only [6]
  2. “However, researches has mainly focused on Al-Zn-Mg 62 [15, 16], Al-Mg [17], Al-Cu-Li [17, 19] and other alloys [20, 21], the ultra-high temperature 63 treatment of Al-Zn-In alloy after the addition of Er has not been reported.”

The high-temperature homogenization treatment of the Er containing Al alloys was investigated as an example of the Al-Cu-Er based alloys.

The comparison with its results is very suitable.

  1. General for Introduction part. The Introduction should be significantly improved to cover more researchers in accordance with the previous comments.
  2. “Master alloys of Al-10 wt.% Zn and Al-10 wt.% Er were purchased”.

What was the reason to use Al-10 wt.% Zn master alloy? Zinc introduce in the Al alloy in the pure state usually.

  1. “It is a typical as-cast microstructure, exhibiting serious dendritic segregation, which 153 requires further heat treatment to reduce dendritic segregation.”

Please provide the experimental results to demonstrate the serious dendritic segregation.

For example, using SEM EDX analysis of the distribution of elements by line crossing the dendritic cell.  

  1. “It is obvious that severe segregation of Er occurred along the Al grain boundaries.”

Uncorrected sentence. It is not segregation of Er. It is the Er rich phase which should be Al3Er in the binary alloy.

  1. In addition to comment 7. A small amount of In and Zn is dissolved in the Al3Er phase what clearly seen from element map distribution and point analysis. It is means that in the investigated alloy forms the (Al,Zn)3(Er,In) phase of the eutectic origin.

This very interesting moment should be investigated in detail.

In addition. The ternary phase formation is possible in the ternary Al-Zn-Er alloys. Please analyze the ternary Al-Zn-Er phase diagram.

  1. “The peak at 614.3 ℃ of the cooling DSC curve possibly corresponds to the melting  point of the quasi-eutectic structure (α(Al)+Al3Er) [13, 23]…”

The [13,23] refs have not investigated the Al alloys with Er addition.

The comparison with the binary Al-Er phase diagram is more reasonable.

  1. The quality of samples for microstructure investigation is very low. There are many scratches and artifacts.

 The quality should be improved.

  1. The composition of the phases in Figs 6 and 8 should be experimentally determined. For example, possible two types of intermetallic phases are visible in Fig 6b and Fig 8c.
  2. “The content of Al3Er phases in the alloy began to decrease as the UHTT temperature increased, and the Al3Er precipitated 233 phase began to dissolve back into the α(Al) matrix. After UHTT at (615 ℃, 24 h), combined with Figure 6e and the Al3Er peaks at 615 ℃, the Al3Er phases were almost completely dissolved back into the α(Al) matrix.”

It are a very doubtful sentence. The Al3Er phase in the Al alloy with 1wt%Er cannot be almost completely dissolved into the α(Al) matrix. The maximum Er solubility is about 0.3wt%.

As can clearly seen from XRD patterns the Al3Er phase peaks are presented at all temperatures in fig 7 and all times in Fig 9.

The Al3Er phase should present in the alloy indented the temperature and time of the annealing in accordance with the phase diagram. And the Al3Er phase may be in two conditions – eutectic and precipitation.

  1. “Figure 13. Evolution of the statistical amount of grain boundary phase after treatment at 615 °C for 374 different times (0 h, 16 h, 24 h, 32 h, 40 h).”

Was the grain boundary phase only analyzed or interdendritic else?

  1. Fig 10 is absolutely uninformative.
  2. Lines 338-358. The TEM results should be supported by images at high magnification. Very important to demonstrate the boundary between Al and precipitate after high-temperature annealing.

A comparison with other researchers’ results is necessary.  

  1. Conclusion 1. No “Serious dendritic segregation” was not demonstrated.

Author Response

Dear Reviewer:

Thank you for your careful review and comments concerning our manuscript entitled “Effect of heat treatment on microstructural evolution and microhardness change of Al-5Zn-0.03In-1Er alloy”. Those comments are all valuable and very helpful for revising and improving our paper, as well as the important guiding significance to our researches. We have studied comments carefully and have made major changes which we hope meet with approval. Revised portion are marked in red in the paper. The responds to your comments are as follows:

Point 1: “Studies have shown that adding an appropriate amount of Er to the Al alloy can form Al3Er phase with the stable L12 structure 38 [3, 4].”

The origin of the Al3Er phase should be indicated.

The Al3Er phase formation in the Al-Er alloy first of all follows from the binary phase diagram.

Al3Er phase formation in the Al alloys was shown significantly earlier.

Please change an approach to site the literature.

Response: We have added the explanation of Al3Er formation. The formation of Al3Er has been indicated in line 38-45:

“Studies have shown that adding an appropriate amount of Er to the Al alloy can form Al3Er phase with the stable L12 structure [3, 4]. According to the Al-Er binary phase diagram [3], there is eutectic Al-Al3Er on the left side of the phase diagram, and the eutectic point composition is about 6% Er. Generally speaking, the content of Er in the experimental alloy is much less than 6%. Zhu, S. et al. [7] found that a small part of Er solid solution forms supersaturated solid solution in the matrix, most of Er is enriched near the interface (beyond the eutectic point component) to form eutectic Al-Al3Er.”

Reference

[7] Zhu, S.; Huang, H.; Nie, Z.; Wen, S.; Zhang, Z. Formation and evolution of Al3Er phrase in Al-Er alloy. Chinese Journal of Rare Metals. 2009, 33(2):164-169, doi: 10.13373/j.cnki.cjrm.2009.02.016.

Point 2: The grain refining effect of Er in the Al alloys are investigated by more researchers than only [6]

Response: Thank you for your suggestions. We have added more references to emphasize the grain refining effect of Er in the Al alloys. The modified content in line 52-55 is as follows:

“Another Al3Er segregation phase produces a large number of pinning dislocations and sub-grain boundaries, which will seriously hinder the movement of dislocations, inhibit the nucleation and growth of crystallization, and further refine grains and strengthen the alloy [6-9].”

Reference

[7] Zhu, S.; Huang, H.; Nie, Z.; Wen, S.; Zhang, Z. Formation and evolution of Al3Er phrase in Al-Er alloy. Chinese Journal of Rare Metals. 2009, 33(2):164-169, doi: 10.13373/j.cnki.cjrm.2009.02.016.

[8] Zhang, X.; Wang, H.; Yan, B.; Zou, C.; Wei, Z. The effect of grain refinement and precipitation strengthening induced by Sc or Er alloying on the mechanical properties of cast Al-Li-Cu-Mg alloys at elevated temperatures. Mater. Sci. Eng. A. 2021,822:141641, doi: 10.1016/j.msea.2021.141641.

[9] Qian, W.; Zhao, Y.; Kai, X.; Gao, X.; Miao, C. Characteristics of microstructural and mechanical evolution in 6111Al alloy containing Al3(Er,Zr) nanoprecipitates. Mater. Charact, 2021, 178:111310, doi: 10.1016/j.matchar.2021.111310.

Point 3: “However, researches has mainly focused on Al-Zn-Mg 62 [15, 16], Al-Mg [17], Al-Cu-Li [17, 19] and other alloys [20, 21], the ultra-high temperature 63 treatment of Al-Zn-In alloy after the addition of Er has not been reported.”

The high-temperature homogenization treatment of the Er containing Al alloys was investigated as an example of the Al-Cu-Er based alloys.

The comparison with its results is very suitable.

Response: Thank you for your suggestions. We have added more references on heat treatment of Er-containing Al alloys to support the novelty of this paper. The revised content and the added literature in line 77-81 are as follows:

“Researchers have done much work on the treatment of aluminum alloys. However, researches has mainly focused on Al-Zn-Mg [22, 23], Al-Mg [24], Al-Cu-Li [25, 26], Al-Mg-Mn-Er [27], Al-Zn-Mg-Cu-Zr-Er [28] and other alloys [29, 30], the ultra-high temperature treatment of Al-Zn-In alloy after the addition of Er has not been reported.”

Reference

[27] Fu, L.; Li, Y.; Jiang, F.; Xu, G.; Yin, Z. On the role of Sc or Er micro-alloying in the microstructure evolution of Al-Mg alloy sheets during annealing. Mater. Charact. 2019, 157:109918, doi: 10.1016/j.matchar.2019.109918.

[28] Wu, H.; Wen, S.; Lu, J.; Mi, Z.; Zeng, X,; Huang, H.; Nie, Z.; Microstructural evolution of new type Al–Zn–Mg–Cu alloy with Er and Zr additions during homogenization. Trans. Nonferrous Met. Soc. China, 2017, 27, 1476-1482, doi: 10.1016/S1003-6326(17)60168-7.

Point 4: General for Introduction part. The Introduction should be significantly improved to cover more researchers in accordance with the previous comments.

Response: Thank you for your reminder. We have added more references in the light of previous reviews. If the introduction is insufficient, we will continue to consult and improve it. Added references is summarized as follows:

[7] Zhu, S.; Huang, H.; Nie, Z.; Wen, S.; Zhang, Z. Formation and evolution of Al3Er phrase in Al-Er alloy. Chinese Journal of Rare Metals. 2009, 33(2):164-169, doi: 10.13373/j.cnki.cjrm.2009.02.016.

[8] Zhang, X.; Wang, H.; Yan, B.; Zou, C.; Wei, Z. The effect of grain refinement and precipitation strengthening induced by Sc or Er alloying on the mechanical properties of cast Al-Li-Cu-Mg alloys at elevated temperatures. Mater. Sci. Eng. A. 2021,822:141641, doi: 10.1016/j.msea.2021.141641.

[9] Qian, W.; Zhao, Y.; Kai, X.; Gao, X.; Miao, C. Characteristics of microstructural and mechanical evolution in 6111Al alloy containing Al3(Er,Zr) nanoprecipitates. Mater. Charact, 2021, 178:111310, doi: 10.1016/j.matchar.2021.111310.

[27] Fu, L.; Li, Y.; Jiang, F.; Xu, G.; Yin, Z. On the role of Sc or Er micro-alloying in the microstructure evolution of Al-Mg alloy sheets during annealing. Mater. Charact. 2019, 157:109918, doi: 10.1016/j.matchar.2019.109918.

[28] Wu, H.; Wen, S.; Lu, J.; Mi, Z.; Zeng, X,; Huang, H.; Nie, Z.; Microstructural evolution of new type Al–Zn–Mg–Cu alloy with Er and Zr additions during homogenization. Trans. Nonferrous Met. Soc. China, 2017, 27, 1476-1482, doi: 10.1016/S1003-6326(17)60168-7.

Point 5: “Master alloys of Al-10 wt.% Zn and Al-10 wt.% Er were purchased”.What was the reason to use Al-10 wt.% Zn master alloy? Zinc introduce in the Al alloy in the pure state usually.

Response: In our previous works, we found that when Zn is directly added into the crucible in the form of pure metal, there are serious burning loss and other problems, such as volatile at high temperature and contamination of crucible. Adding Al melt in the form of Al-Zn master alloy can effectively avoid the occurrence of the above problems and effectively reduce the loss of Zn in the process of alloy melting.

Point 6: “It is a typical as-cast microstructure, exhibiting serious dendritic segregation, which 153 requires further heat treatment to reduce dendritic segregation.”

Please provide the experimental results to demonstrate the serious dendritic segregation.

For example, using SEM EDX analysis of the distribution of elements by line crossing the dendritic cell.  

Response: We have studied the microstructure of Al-5Zn-0.03In-xE alloy before [1], and confirmed that the precipitated phase is basically interdendritic phase from the microstructure and morphology characteristics in Figure 1. As shown in Figure 1a, there are obvious dendrites and the interdendritic phase in the SEM image(Figure 1b) is also very obvious. Therefore, no repetitive work has been done in this article.

(b)

(a)

(d)

(c)

Figure 1. OM and SEM photographs of anode alloys: (a)OM of Al-5Zn-0.03In-0.4Er; (b)SEM of Al-5Zn-0.03In-0.4Er; (c) OM of Al-5Zn-0.03In-1Er; (d) SEM of Al-5Zn-0.03In-1Er.

 Combined with our previous work [14], dendritic segregation phase was clearly seen in the Al-5Zn-0.03In alloy containing Er in Figure 2. With the increase of Er element content, the dendrite morphology changed greatly, from coarse to long strip, fine equiaxed; the number of interdendritic precipitates gradually increased, and the morphology gradually changed from intermittent to continuous network. Therefore, the SEM morphology of Al-5Zn-0.03In-1Er alloy selected in this paper is generally judged to be interdendritic segregation.

Figure 2. Dendrites of aluminum alloy: (a) Al-5Zn-0.03In, (b) Al-5Zn-0.03In-1Er, (c) Al-5Zn-0.03In-4Er, and (d) Al-5Zbn-0.03In-7Er

Reference:

[1] Li, H. Effect of Er on Microstructure and Properties of Al-Zn-In anode [D]. Guangxi University. 2016: 31-33.

[14] Li, H.; Wei, B.; Xu, Z.B.; Zeng, J.M.; Chen, R.; Li, H.; Lu, Y.Y. Effect of Er on Microstructure and Electrochemical Performance of Al-Zn-In Anode. Rare Metal Mat. Eng. 2016, 45, 1848-1854, doi: CNKI:SUN:COSE.0.2016-07-040.

Point 7: “It is obvious that severe segregation of Er occurred along the Al grain boundaries.”

Uncorrected sentence. It is not segregation of Er. It is the Er rich phase which should be Al3Er in the binary alloy.

Response: Thank you for your suggestions. We have modified the “severe segregation of Er” to “Er-rich phase” in line 178-179:

“The corresponding distributions of Al, Zn, In, and Er are shown in Figure 4b-e. It is obvious that Er-rich phase occurred along the Al grain boundaries.”

Point 8: In addition to comment 7. A small amount of In and Zn is dissolved in the Al3Er phase what clearly seen from element map distribution and point analysis. It is means that in the investigated alloy forms the (Al,Zn)3(Er,In) phase of the eutectic origin.

This very interesting moment should be investigated in detail.

In addition. The ternary phase formation is possible in the ternary Al-Zn-Er alloys. Please analyze the ternary Al-Zn-Er phase diagram.

Response: Because the little content of In, it will not be considered first. It can be obtained from Al-Zn-Er ternary phase diagram, The composition point of Al-5Zn-1Er falls in the Al-rich angle of the ternary phase diagram. The products formed are basically eutectic Al -Al3Er. It may be accompanied by a small amount of ErZn5Al3 that is difficult to detect by XRD.

Al3Er+(Al)

Al-Zn-Er ternary phase diagram at 450℃ (dotted line probable, dashed line predicted)

Reference

 Ding, Jiating, Zhu, et al. Isothermal Section of the Al-Zn-RE (RE = Ho, Er) Systems at 450 ℃. J. phase equilib.diff, 37(6):658-663, doi: 10.1007/s11669-016-0494-7.

Point 9: “The peak at 614.3 ℃ of the cooling DSC curve possibly corresponds to the melting  point of the quasi-eutectic structure (α(Al)+Al3Er) [13, 23]…”

The [13,23] refs have not investigated the Al alloys with Er addition.

The comparison with the binary Al-Er phase diagram is more reasonable.

Response: According to Al-Er binary phase diagram [3], When the Er content is 1 wt.%, there is eutectic Al-Al3Er on the left side of the phase diagram, the onset point in the DSC curve may correspond to the eutectic point according to the relevant references [20,30,31,32]. The modified content in line 198-202 is as follows:

“According to Al-Er binary phase diagram [3], the peak at 614.3 ℃ of the cooling DSC curve possibly corresponds to the melting point of the quasi-eutectic structure (α(Al)+Al3Er) [20], which is similar to general Al-Cu alloy where the DSC endothermic peak at 532 ℃, is largely caused by the melting of the Al2Cu phase [32-34], and the onset at 627.7 ℃ is related to the initial melting point of the α(Al) matrix [35].”

Reference

[20] Li, S.; He, C.; Fu, J.; Xu, J.; Wang, Z. Evolution of microstructure and properties of novel aluminum‑lithium alloy with different roll casting process parameters during twin-roll casting. Mater. Charact. 2020, 161:110145, doi: 10.1016/j.matchar.2020.110145.

Point 10: The quality of samples for microstructure investigation is very low. There are many scratches and artifacts.

 The quality should be improved.

Response: Due to the epidemic, we are unable to go back to school to prepare samples and conduct experiments, which we will make up for it later if necessary.

Point 11: The composition of the phases in Figs 6 and 8 should be experimentally determined. For example, possible two types of intermetallic phases are visible in Fig 6b and Fig 8c.

Response: Combined with the scan analysis of the as-cast Al-5Zn-0.03In-1Er alloy and after treatment at 615 °C for different time in Figure 10, we can conclude that the segregation of Er exists. Combined with our previous work [14], the second phase is basically interdendritic phase or residual segregation phase after dissolved in the matrix.

Point 12: “The content of Al3Er phases in the alloy began to decrease as the UHTT temperature increased, and the Al3Er precipitated 233 phase began to dissolve back into the α(Al) matrix. After UHTT at (615 ℃, 24 h), combined with Figure 6e and the Al3Er peaks at 615 ℃, the Al3Er phases were almost completely dissolved back into the α(Al) matrix.”

It are a very doubtful sentence. The Al3Er phase in the Al alloy with 1wt%Er cannot be almost completely dissolved into the α(Al) matrix. The maximum Er solubility is about 0.3wt%.

As can clearly seen from XRD patterns the Al3Er phase peaks are presented at all temperatures in fig 7 and all times in Fig 9.

The Al3Er phase should present in the alloy indented the temperature and time of the annealing in accordance with the phase diagram. And the Al3Er phase may be in two conditions – eutectic and precipitation.

Response: Thank you for your reminder. Combined with the evolution of the second phase in Fig 6, we found that Al3Er phase exists in different heat treatment temperatures and Al3Er phase. The Al3Er phase cannot be almost completely dissolved into the α(Al) matrix. So, we have modified the analysis in line 245-247:

“After UHTT at (615 ℃, 24 h), combined with Figure 6e and the Al3Er peaks at 615 ℃, a quantity of Al3Er phase was dissolved back into the α(Al) matrix.”

Point 13: “Figure 13. Evolution of the statistical amount of grain boundary phase after treatment at 615 °C for 374 different times (0 h, 16 h, 24 h, 32 h, 40 h).”

Was the grain boundary phase only analyzed or interdendritic else?

Response: Combined with the response to point 6, we have modified “grain boundary phase” to “interdendritic” to describe it in a more comprehensive way.

Point 14: Fig 10 is absolutely uninformative.

Response: We have modified the discussion of the line scanning analysis to highlight the purpose of research: the segregation of Er exists in ac-cast alloy and alloy treated for different treatment time at 615℃. The revised content in line 289-302 is as follows:

“To explore the change of micro-segregation in the as-cast and treated alloys, energy-dispersive spectrometer (EDS, OXFORD) linear scanning was adopted and the result is shown in Figure 10. The yellow line in the image expresses the scanning position. The result of the ac-cast alloy is shown in Fig 10(a), while Fig 10(b-e) show the result of the alloy treated at 615 ℃ for 16h, 24h, 32h and 40h. The corresponding distribution results of the main alloying elements are shown on the image. Obviously, there is almost no obvious change in the segregation of the main elements between the as-cast alloy and the alloy UHTT at 615 ℃ for different time. The segregation of In is less evident because of its lower content, and the segregation of In can be neglected. The Zn element distribution is homogeneous from the grain boundary to the inside. However, the concentration of Er content in the segregation phase is significantly higher than that in grain interior, and this result indicates the existence of Er segrega-tion. This might be due to the low diffusion coefficient of Er element [18]. Therefore, we can conclude that the diffusion velocity of Zn is faster than that of Er and the segregation of Er always exists.”

Point 15: Lines 338-358. The TEM results should be supported by images at high magnification. Very important to demonstrate the boundary between Al and precipitate after high-temperature annealing.

A comparison with other researchers’ results is necessary.  

Response: Thank you for your suggestions. We have added to the high-resolution transmission of the boundary between Al and precipitate after treatment. The supplementary content is as follows:

“The Al–5Zn–0.03In–1Er alloy treated at 615 ℃ for 32h was characterized by TEM to further determine the composition of the precipitated phase in Figure 12. The TEM image of the alloy is shown in Figure 12a and the high-resolution TEM characterization of both sides of the boundary is shown in Figure 12b. Figure 12b shows that the second phase is on the left side of the boundary, the matrix is on the right side, and results analyzed by Digital Micrograph software. After performing the Fourier transform of the image and measuring the distance, the distance between the crystal planes of second phase is about 0.421 nm, which accords with the distance between the crystal planes of Al3Er, and the distance between the crystal planes of matrix is about 0.199 nm, which satisfies the distance between the crystal planes of Al-Zn solid solution. The selected area electron diffraction patterns (SAEDP) of the second phase and the matrix were shown in Figure12c and Figure12d. After calibrating the SAEDPs, the result is the same as the crystal face indices of the Al3Er and Al–Zn solid solution.”

(b)

(a)

(d)

(c)

Figure 12. TEM images of the Al-5Zn-0.03In-1Er alloy treated at 615 ℃for 32 h: (a)TEM image,
(b) HRTEM, (e) the SAEDP of the second phase, (f) the SAEDP of the alloy matrix.

Point 16: Conclusion 1. No “Serious dendritic segregation” was not demonstrated.

Response: According to our previous work [14], we thought that the “Serious dendritic segregation” is not accurate, so we modified “Serious dendritic segregation” to “interdendritic segregation” in line 445-448:

“The interdendritic segregation exists in the as-cast Al-Zn-In-Er alloy. With the in-crease of UHTT time, the main element Er is largely enriched in grain boundaries, and its concentration decreases from the grain boundary to the inside, but the changes of Zn, In are not obvious”.

Reviewer 2 Report

The paper is focused on the microstructure and thermal analysis of as-cast Al-5Zn-0.03In-1Er. The topic falls within the scope of the journal. The presentation and discussion of the results could be improved. I recommend the publication after the following revisions:

  • DSC curve (Fig. 5). I suggest to determine the enthalpy changes of the melting processes by the integration of the endothermic peaks.
  • Experimental details for SEM analyses (energy of beam, working distance,..) are missed. Please add them. Similarly, I recommend to detail the experimental conditions used for XRD measurements.
  • Figure 10. The presentation and discussion of the line scanning analysis should be improved.
  • Figure 11. How do you estimate the error for the hardness values?

Author Response

Dear Reviewer:

Thank you for your careful review and comments concerning our manuscript entitled “Effect of heat treatment on microstructural evolution and microhardness change of Al-5Zn-0.03In-1Er alloy”. Those comments are all valuable and very helpful for revising and improving our paper, as well as the important guiding significance to our researches. We have studied comments carefully and have made major changes which we hope meet with approval. Revised portion are marked in red in the paper. The responds to your comments are as follows:

Point 1: DSC curve (Fig. 5). I suggest to determine the enthalpy changes of the melting processes by the integration of the endothermic peaks.

Response: Many studies on heat treatment for homogenization have use DSC curve to explore the evolution of the phases. For example, ChunMei Li [1] used DSC to find the melting points of α-η(MgZn2) and α-S(Al2CuMg) to explore the evolution of the phases. Wen Kai a [2] used DSC to determine the melting temperature of Mg(Zn, Cu, Al)2 phase to investigate the microstructural evolution of a high Zn-containing Al-Zn-Mg-Cu alloy during homogenization. So it’s necessary for us to determine the melting points of specific phase and solidus temperature of the alloy and the enthalpy changes is not necessary. Thank you for your suggestions.

References

[1] ChunMei, Li, ZhiQian, et al. Intermetallic phase formation and evolution during homogenization and solution in Al-Zn-Mg-Cu alloys[J]. Science China Technological Sciences, 2013.

[2] A W K ,  A X B ,  A Z Y , et al. Microstructure Evolution of a High Zinc Containing Al-Zn-Mg-Cu Alloy during Homogenization[J]. Rare Metal Materials and Engineering, 2017, 46( 4):928-934.

Point 2: Experimental details for SEM analyses (energy of beam, working distance,..) are missed. Please add them. Similarly, I recommend to detail the experimental conditions used for XRD measurements.

Response: The energy of beam, working distance, model type of scanning electron microscope and magnification of images were added in all SEM images. For example, figure 3b below shows main experimental details for SEM analysis.

(b)

(a)

SU8020 15.0kV 11.3mm ×350 SE(UL)

Figure 3. Microstructure of the as-cast Al-5Zn-0.03In-1Er alloy: (a) OM image; (b) SEM image.

The experimental details of XRD measurement were added in the section 2.3 “Microhardness measurements”. The modified content in line123-125 is as follows:

“Alloy phases were identified by using a X-ray diffraction (XRD, Rigaku D2500 V) operating at 40kV and 200mA with Cu Kα radiation. The scanning range was between 20° and 90° (2θ), with a scanning rate of 4°/min.”

Point 3: Figure 10. The presentation and discussion of the line scanning analysis should be improved.

Response: We have modified the discussion of the line scanning analysis to highlight the purpose of research: the segregation of Er exists in ac-cast alloy and alloy treated for different treatment time at 615℃. The revised content in line 289-302 is as follows:

“To explore the change of micro-segregation in the as-cast and treated alloys, energy-dispersive spectrometer (EDS, OXFORD) linear scanning was adopted and the result is shown in Figure 10. The yellow line in the image expresses the scanning position. The result of the ac-cast alloy is shown in Fig 10(a), while Fig 10(b-e) show the result of the alloy treated at 615 ℃ for 16h, 24h, 32h and 40h. The corresponding distribution results of the main alloying elements are shown on the image. Obviously, there is almost no obvious change in the segregation of the main elements between the as-cast alloy and the alloy UHTT at 615 ℃ for different time. The segregation of In is less evident because of its lower content, and the segregation of In can be neglected. The Zn element distribution is homogeneous from the grain boundary to the inside. However, the concentration of Er content in the segregation phase is significantly higher than that in grain interior, and this result indicates the existence of Er segrega-tion. This might be due to the low diffusion coefficient of Er element [18]. Therefore, we can conclude that the diffusion velocity of Zn is faster than that of Er and the segregation of Er always exists.”

Point 4: Figure 11. How do you estimate the error for the hardness values?

Response: We used the average of microhardness values to represent the concentrated trend of microhardness of each sample. When there are two extreme values (i.e., maximum and minimum) in each group of values, the representative effect of the average of microhardness values of this group will be weakened. Therefore, in order to eliminate this phenomenon, we can remove the two extreme values, calculate only the average of the remaining three values, and take the results as the average of all values. At the same time, the error of hardness value can be reduced. And we used the standard deviation of each set of microhardness values to make the error bar in figure 11 to represent the error range.

Reviewer 3 Report

It is the resubmsission of the manuscript metals-1508868.

As already evidenced, in my opinion it can be accepted in the present form

Author Response

Dear Reviewer:

Thank you for your careful review and comments concerning our manuscript entitled “Effect of heat treatment on microstructural evolution and microhardness change of Al-5Zn-0.03In-1Er alloy”. Best wishes to you.

Reviewer 4 Report

The authors propose a special heat treatment for the homogenization of the Al-Zn-In-Er alloy. The authors have considered every aspect dealing with the success of this treatment, and a systematic and comprehensive research on this topic has been carried out. Besides, results in abundance, and the organization of this paper is also properly arranged, and easily understood. However, the points listed below cause me concern.

1) The authors focus on the study of the Al-Zn-In-Er alloy, but the content of indium is so low that the authors neglect it throughout the entire study, since they do not even “see” it. This point needs to be discussed. Please explain the need for such a small amount of indium.

2) The authors write “The highest and lowest values in each group of data were omitted, and the average of the remaining three microhardness values was taken as the microhardness value of the alloy.” This approach seems strange. Usually either the results of all measurements are taken into account, or their statistical evaluation is carried out using known criteria

3) In fact, the microhardness values for 32 and 40 hours in Figure 11 are not statistically significant. Therefore, it is incorrect to say that the microhardness decreases.

4) In addition, there are typos, like replacing a dot with a comma, grammatical errors, etc.

Author Response

Dear Reviewer:

Thank you for your careful review and comments concerning our manuscript entitled “Effect of heat treatment on microstructural evolution and microhardness change of Al-5Zn-0.03In-1Er alloy”. Those comments are all valuable and very helpful for revising and improving our paper, as well as the important guiding significance to our researches. We have studied comments carefully and have made major changes which we hope meet with approval. Revised portion are marked in red in the paper. The responds to your comments are as follows:

Point 1: The authors focus on the study of the Al-Zn-In-Er alloy, but the content of indium is so low that the authors neglect it throughout the entire study, since they do not even “see” it. This point needs to be discussed. Please explain the need for such a small amount of indium.

Response: Thank you for your suggestion. We have added the explanation for the low content of In and the effect of In on Al-based anode. The additions in line 58-65 are as follows:

“Li Y. et al. [13] studied the effect of In content on Al-Zn-In-Mg-Ti anodes and con-firmed that high In content will lead to uneven dissolution morphology and decrease current efficiency. Therefore, the content of In in aluminum-based anode alloy is gen-erally less than 0.03 wt.%. Due to the low solid solubility of In in Al-base alloy, In ele-ment mainly exists in the form of segregation phase. Although the In content is little, it can also play a good activation role on Al-based anode. The anodes containing In without heat treatment can still achieve high current efficiency. Meanwhile, Al-Zn-In alloys have some disadvantages, such as uneven surface dissolution, poor mechanical properties and so on.”

Reference

[13] Li, Y.; Wen, J.; Zhao, S.; Ma, J. Effect of In Content On Microstructure and Electrochemical Performance of Al-Zn-In-Mg-Ti Alloy. Corros. Sci. & Prot. Technol, 2010,3(22): 216-219, doi: 10.1155/2010/293410.

Point 2: The authors write “The highest and lowest values in each group of data were omitted, and the average of the remaining three microhardness values was taken as the microhardness value of the alloy.” This approach seems strange. Usually either the results of all measurements are taken into account, or their statistical evaluation is carried out using known criteria.

Response: We used the average of microhardness values to represent the concentrated trend of microhardness of each sample. When there are two extreme values (i.e., maximum and minimum) in each group of values, the representative effect of the average of microhardness values of this group will be weakened. Therefore, in order to eliminate this phenomenon, we can remove the two extreme values, calculate only the average of the remaining three values, and take the results as the average of all values. At the same time, the error of hardness value can be reduced.

Point 3: In fact, the microhardness values for 32 and 40 hours in Figure 11 are not statistically significant. Therefore, it is incorrect to say that the microhardness decreases.

Response: Thank you for your suggestions. This is really a mistake of our expression. As explained in the section “discussion”, when Al-5Zn-0.03In-1Er alloy was treated at 615℃ for 32h, the anchoring of dislocations due to precipitation phases (Al3Er) can hinder the slip of dislocations. This is a main cause of the higher microhardness of the alloy after UHTT. When the heat treatment time exceeds 32h, the phenomenon of anchoring dislocation caused by precipitation phase (Al3Er) still exists, and the influence caused by “anchoring dislocation” on microhardness will not change significantly. So, we have modified the expression as follows in line 310-312:

“Figure 11 shows that the microhardness first increased to the maximum when the treatment was 32 h and then hardly changed with increasing treatment time.”

Point 4: In addition, there are typos, like replacing a dot with a comma, grammatical errors, etc.

Response: We have checked the whole paper and corrected the grammatical and symbolic errors. All in all, thank you for your suggestions.

Reviewer 5 Report

 - Please add the DOI links for all references.

- Add more current references (2020 – 2021).

- Highlight better which is the novelty of the work?

-  Specify where the raw materials were purchased.

- What is the status of the literature according to your work? Make a comparison between the results obtained by you and another previous research.

- Need to actually explain how the EDS % composition values were obtained (spectra or area scan? number of spectra obtained to get the statistics?).

- Please discuss in Introduction about the applications for the Al-Zn-In-Er alloy.

- Complete the conclusions with the limitations of the proposed methodology. Also write future research.

- Generally the quality of the writing could be improved.

Author Response

Point 1: Please add the DOI links for all references.

Response: Thank you for your review. We have added all doi links for all references.

Point 2: Add more current references (2020 – 2021).

Response: We have added more up-to-date references which are concerned with grain refinement of Er and application fields of Al-Zn-In-Er alloys respectively in the introduction. The supplemented references are as follows:

References

[8] Zhang, X.; Wang, H.; Yan, B.; Zou, C.; Wei, Z. The effect of grain refinement and precipitation strengthening induced by Sc or Er alloying on the mechanical properties of cast Al-Li-Cu-Mg alloys at elevated temperatures. Mater. Sci. Eng. A. 2021,822:141641, doi: 10.1016/j.msea.2021.141641.

[9] Qian, W.; Zhao, Y.; Kai, X.; Gao, X.; Miao, C. Characteristics of microstructural and mechanical evolution in 6111Al alloy containing Al3(Er,Zr) nanoprecipitates. Mater. Charact, 2021, 178:111310, doi: 10.1016/j.matchar.2021.111310.

[10] Wang, H.; Min, D.; Liang, H.; Gao, Q. Study on Al-Zn-In Alloy as Sacrificial Anodes in Seawater Environment. J. Ocean. U. China. 2019.18(04):889-895, doi: CNKI:SUN:QDHB.0.2019-04-014.

[12] Wu, Z.; Zhang, H.; Yang, D.; Zou, J.; Nagaumi, H.; Electrochemical behaviour and discharge characteristics of an Al–Zn–In–Sn anode for Al-air batteries in an alkaline electrolyte. J. Alloys Compd, 2020, 837:155599, doi: 10.1016/j.jallcom.2020.155599.”

[20] Li, S.; He, C.; Fu, J.; Xu, J.; Wang, Z. Evolution of microstructure and properties of novel aluminum‑lithium alloy with different roll casting process parameters during twin-roll casting. Mater. Charact. 2020, 161:110145, doi: 10.1016/j.matchar.2020.110145.

Point 3: Highlight better which is the novelty of the work?

Response: At present, there is no research report on the effect of utra-high temperature homogenization on the microstructure and properties of Al-Zn-In anode alloy with Er addition. We added the novelty of the article in the abstract in line 12-13:

“However, Er segregation in solid solutions which reduces the comprehensive properties of alloys is difficult to reduce and there is no report on the homogenization of Al-Zn-In alloys.”

and emphasized the novelty of the article in 77-81:

“Researchers have done much work on the treatment of aluminum alloys. However, re-searches has mainly focused on Al-Zn-Mg [22, 23], Al-Mg [24], Al-Cu-Li [25, 26], Al-Mg-Mn-Er [27], Al-Zn-Mg-Cu-Zr-Er [28] and other alloys [29, 30], the ultra-high temperature treatment of Al-Zn-In alloy after the addition of Er has not been reported.”

Point 4: Specify where the raw materials were purchased.

Response: Raw materials were commercial pure Al, pure In, master alloys of Al-10 wt.% Zn and Al-10 wt.% Er purchased from Hunan rare Earth Institute.

Point 5: What is the status of the literature according to your work? Make a comparison between the results obtained by you and another previous research.

Response: Our previous work is mainly to explore the effect of different Er content on the electrochemical behaviours of Al-5Zn-0.03In-xEr alloy [14,15]. And we found when Er content was about 1%, the current efficiency of the alloy was approximately highest, the grain of the alloy was refined and the character of "uniform corrosion" was exhibited.

Afterwards, we found that high temperature heat treatment can significantly reduce the segregation phase in the alloy, making the alloy more uniform. By exploring the microstructure evolution of Al-5Zn-0.03In-1Er with change of temperature and time at high temperature, supplemented by the change of microhardness, we can explore and obtain that the appropriate homogenization system of Al-5Zn-0.03In-1Er is 615 ℃×32h. We can obtain more uniform structure and improve the current efficiency of the alloy through the appropriate homogenization system, which meets the demand of improving the current efficiency as much as possible in our previous work.

[14] Li, H.; Wei, B.; Xu, Z.B.; Zeng, J.M.; Chen, R.; Li, H.; Lu, Y.Y. Effect of Er on Microstructure and Electrochemical Performance of Al-Zn-In Anode. Rare Metal Mat. Eng. 2016, 45, 1848-1854, doi: CNKI:SUN:COSE.0.2016-07-040.

[15] Shen, Z.N.; Chen, X.Y.; Li, H.; Xu, Z.B.; Zeng, J.M. Effect of Er Content on the Microstructure and Current Efficiency of Al-Zn-In-xEr Alloy. Foundry Technol. 2018, 39, 765-768, doi: 10.16410/j.issn1000-8365.2018.04.006.

Point 6: Need to actually explain how the EDS % composition values were obtained (spectra or area scan? number of spectra obtained to get the statistics?).

Response: We used Energy Dispersive Spectrometer point scan to test the type and content of elements in a certain position of the material of point A and point B in Figure 4. So, we used energy spectrum for point scan. Energy dispersive spectroscopy (EDS) combined with scanning electron microscope can be used to analyze the type and content of elements in the micro area of the material. The working principle is that: various elements have their own X-ray characteristic wavelengths, and the size of the characteristic wavelengths depends on the characteristic energy E released during the energy level transition. the energy spectrometer uses the characteristic energy of X-ray photons of different elements to carry out composition analysis. The accuracy of energy spectrum quantitative analysis is related to the sample preparation process, the conductivity of the sample, the content of elements and the atomic number of elements. 

And we have added to the high-resolution transmission of the boundary between Al and precipitate after treatment to determine the composition of the precipitated phase and matrix in the alloy in line 184-188:

“Combined with the TEM characterization of the following Figure 12, 13, EDS analysis reveals that the as-cast alloy is mainly composed of dendritic α-Al matrix, as marked as point A and the Al3Er phase within the α-Al matrix [31], as marked as point B in Figure 4a.”

Point 7: Please discuss in Introduction about the applications for the Al-Zn-In-Er alloy.

Response: Al-Zn-In alloys are widely used in the field of corrosion as anode alloys. We reviewed the latest literature and supplemented the main application fields of the alloy in line 55-57.

“Al-Zn-In alloys are favored by researchers because of their excellent electrochemical performance, and are widely used in the field of corrosion as anode alloys, such as seawater [10], deep water [11] and alkaline electrolyte [12].”

References

[10] Wang, H.; Min, D.; Liang, H.; Gao, Q. Study on Al-Zn-In Alloy as Sacrificial Anodes in Seawater Environment. J. Ocean. U. China. 2019.18(04):889-895, doi: CNKI:SUN:QDHB.0.2019-04-014.

[11] Sun, H.; Liu, L.; Li, Y.; Ma, L.; Yan, Y. The performance of Al–Zn–In–Mg–Ti sacrificial anode in simulated deep water environment. Corros. Sci. 2013, 77, 77-87, doi: 10.1016/j.corsci.2013.07.029.

[12] Wu, Z.; Zhang, H.; Yang, D.; Zou, J.; Nagaumi, H.; Electrochemical behaviour and discharge characteristics of an Al–Zn–In–Sn anode for Al-air batteries in an alkaline electrolyte. J. Alloys Compd, 2020, 837:155599, doi: 10.1016/j.jallcom.2020.155599.”

Point 8: Complete the conclusions with the limitations of the proposed methodology. Also write future research.

Response: We supplemented our prospects for future research in section 6.

“6. Future research in line 463-470:

“In addition to the above conclusions, this work has the following prospects for the new space cage-like structure: After a preliminary judgment, the space cage-like structure not fully consistent the characteristic of intergranular of a eutectic-like structure. Besides, the spinodal decomposition mechanism can’t explain its transformation mechanism completely. Therefore, it is necessary to conduct further researches that include the following major aspects: (i) the space cage-like structure formation and growth mechanism, (ii) the specific ways of nucleation, growth, impingement of this kind of solid-state phase transformation.”

Point 9: Generally the quality of the writing could be improved.

Response: We have checked the whole paper and corrected the grammatical and symbolic errors. All in all, thank you for your suggestions.

Round 2

Reviewer 1 Report

The Paper was revised but still needs to improve.

  1. Introduction. List of ref contains in the main part from own country researchers. For example, the investigation of S.M. Amer et.al about Al-Cu-Er-based alloys was not analyzed.
  2. About “the serious dendritic segregation”. Incorrect terminology. Erbium leads to the formation of the intermetallic phases and is dissolved in the Al solid solution. Er homogenously distributed in the Al solid solution in accordance with the Author's provided data.
  3. In the Reviewer's opinion, the SEM images with so low quality cannot be published elsewhere. 
  4. Fig 12. The lattice parameter of Al is about 0.4nm. In the obtained results is twice lower. Please describe.
  5. Fig 12. What is the origin of the Al3Er phase? In accordance with the size, it is solidification origin particles. What was the reason for so precise investigation?
  6. In addition to 5. Previously discussed precipitates in Fig 13 have a size of 10-50nm. These particles deserve attention.
  7. The initial phase composition and changes during heat treatment still needed deep analysis.

Author Response

Dear Reviewer:

Thank you for your careful review and comments concerning our manuscript entitled “Effect of heat treatment on microstructural evolution and microhardness change of Al-5Zn-0.03In-1Er alloy”. Those comments are all valuable and very helpful for revising and improving our paper, as well as the important guiding significance to our researches. We have studied comments carefully and have made major changes which we hope meet with approval. Revised portion are marked in red in the paper. The responds to your comments are as follows:

Point 1: Introduction. List of ref contains in the main part from own country researchers. For example, the investigation of S.M. Amer et.al about Al-Cu-Er-based alloys was not analyzed.

Response:We have added more other references on heat treatment of Er-containing Al alloys to support the novelty of this paper, for example, Al-Cu-Er-based alloy studied by S.M. Amer et al. and Al-Er-Zr-based alloy studied by L. Michal et al. The modified content in line 77-80 is as follows:

“However, researches has mainly focused on Al-Zn-Mg [22, 23], Al-Mg [24], Al-Cu-Li [25, 26], Al-Mg-Mn-Er [27], Al-Zn-Mg-Cu-Zr-Er [28], Al-Cu-Er-Mn-Zr[29] and other alloys [30-33], the ultra-high temperature treatment of Al-Zn-In alloy after the addi-tion of Er has not been reported.”

Reference

[29] Amer, S.; Yakovtseva, O.; Loginova, I.; Medvedeva, S.; Prosviryakov, A.; Bazlov, A.; Barkov, R.; Pozdniakov, A. The Phase Composition and Mechanical Properties of the Novel Precipitation-Strengthening Al-Cu-Er-Mn-Zr Alloy. Applied Sciences. 2020, 10(15): 5345, doi: 10.3390/app10155345.

[30] Michal, L.; Martin, V.; Veronika, K.; Hana, K.; Jozef.; Sebastien, Z.; Jakub, C.; Oksana, M.;  František, L. Effect of deformation on evolution of Al3(Er,Zr) precipitates in Al–Er–Zr-based alloy. Mater. Charact. 2022, 186: 111781, doi: 10.1016/j.matchar.2022.111781.

[31] Buranova, Y.; Kulitskiy, V.; Peterlechner, M.; Mogucheva, A,Kaibyshev, R.; Divinski, S.; Wilde, G. Al3(Sc,Zr)-based precipi-tates in Al–Mg alloy: Effect of severe deformation. Acta Mater. 2017, 124, 210-224, doi: 10.1016/j.actamat.2016.10.064.

Point 2: About “the serious dendritic segregation”. Incorrect terminology. Erbium leads to the formation of the intermetallic phases and is dissolved in the Al solid solution. Er homogenously distributed in the Al solid solution in accordance with the Author's provided data.

Response: We have modified “dendritic” to “interdendritic”. We have studied the microstructure of Al-5Zn-0.03In-xE alloy before [1], and confirmed that the precipitated phase is basically interdendritic phase from the microstructure and morphology characteristics in Figure 1. As shown in Figure 1a, there are obvious dendrites and the interdendritic phase in the SEM image(Figure 1b) is also very obvious. Therefore, no repetitive work has been done in this article.

(b)

(a)

(d)

(c)

Figure 1. OM and SEM photographs of anode alloys: (a)OM of Al-5Zn-0.03In-0.4Er; (b)SEM of Al-5Zn-0.03In-0.4Er; (c) OM of Al-5Zn-0.03In-1Er; (d) SEM of Al-5Zn-0.03In-1Er.

Point 3: In the Reviewer's opinion, the SEM images with so low quality cannot be published elsewhere.

Response: We understood that the quality of SEM images is not well and we will make up for the SEM images of better quality in 10 days if necessary.

Point 4: Fig 12. The lattice parameter of Al is about 0.4nm. In the obtained results is twice lower. Please describe.

Response: Thank you for your reminding. The distance between different (hkl) crystal planes (that is, the distance between two adjacent parallel planes) in the same crystal cell is different. We used Digital Micrograph (DM) software to measure the distance between the crystal planes of matrix is about 0.199 nm in Figure 12b, then we concluded that the matrix is Al-Zn solid solution and the Miller indices is “(012)” by comparing with PDF#19-0057 card. We have added Miller index to the discussion to describe the process of characterizing phases more accurately. The modified content in line 357-361 is as follows:

“After performing the Fourier transform of the image and measuring the distance, the distance between the crystal planes of (100) of second phase is about 0.421 nm, which accords with the distance between the crystal planes of Al3Er, and the distance between the crystal planes of (012) of matrix is about 0.199 nm, which satisfies the distance between the crystal planes of Al-Zn solid solution.”

Figure 12b. HRTEM image of the Al-5Zn-0.03In-1Er alloy treated at 615 ℃ for 32 h

Point 5: Fig 12. What is the origin of the Al3Er phase? In accordance with the size, it is solidification origin particles. What was the reason for so precise investigation?

Response: The Al3Er phase in Figure 12 is formed by the fracture of the primary Al3Er phase in the as-cast alloy in Figure 3b after heat treatment at 640 ℃ for 32 h. We have added the explanation of primary Al3Er formation. The formation of Al3Er has been indicated in line 43-46:

“Zhu, S. et al. [7] found that a small part of Er solid solution forms supersaturated solid solution in the matrix, most of Er is enriched near the interface (beyond the eutectic point component), which increases the content of Er near the interface. When the alloy reaches eutectic composition, the eutectic structure of α (Al) and Al3Er is obtained.”

We characterized the alloy treated at 615 ℃ for 32h to further investigate the composition of the precipitated phase. The Al3Er particles decomposed by supersaturated solid solution has good interface compatibility with the matrix, and can form pinning effect to hinder the movement of dislocations, thus achieving the effect of strengthening the alloy. This is why we explored the particle phase.

Reference

[7] Zhu, S.; Huang, H.; Nie, Z.; Wen, S.; Zhang, Z. Formation and evolution of Al3Er phrase in Al-Er alloy. Chinese Journal of Rare Metals. 2009, 33(2):164-169, doi: 10.13373/j.cnki.cjrm.2009.02.016.

Point 6: In addition to 5. Previously discussed precipitates in Fig 13 have a size of 10-50nm. These particles deserve attention.

Response: We have discussed the particles in Figure 13. The Al3Er particles decomposed by supersaturated solid solution has good interface compatibility with the matrix, and can form pinning effect to hinder the movement of dislocations, thus achieving the effect of strengthening the alloy. This is why we explored the particle phase. In this paper, we mainly discuss the strengthening effect of Al3Er particles on the Al-5Zn-0.03In-1Er alloy after high temperature heat treatment, so as to verify that the heat treatment of 615℃×32h is more proper to homogenize the alloy.

Point 7: The initial phase composition and changes during heat treatment still needed deep analysis.

Response: Through XRD patterns in Figure 2, Figure 8 and Figure 9, we found that the the composition phases of as-cast and heat-treated alloys are mainly composed of Al3Er phase and α(Al) phase (Al-Zn solid solution). In figure 12, the Al-5Zn-0.03In-1Er alloy treated at 615 ℃ for 32 h was characterized by TEM, and it was further confirmed that the composition phase was Al3Er phase and α(Al) phase (Al-Zn solid solution). We may investigate more trace and undetected phases in later studies. All in all, thank you for your suggestions.

Figure 2. XRD pattern of the as-cast Al-5Zn-0.03In-1Er alloy.

Figure 7. XRD patterns of Al-5Zn-0.03In-1Er alloy after treatment at different temperatures for 24 h.

Figure 9. XRD patterns of Al-5Zn-0.03In-1Er alloy by treatment at 615 °C for different time.

Reviewer 2 Report

The paper can be accepted in the present form.

Author Response

(The authors gave the same response as above.)

Reviewer 4 Report

After revision, the paper has been improved. However, after making changes to the manuscript, the authors should check the English again. There are new typos. For example, line 435, instead of 'microstructure' it should be 'microstructural'. After that, the paper can be accepted.

Author Response

Dear Reviewer:

Thank you for your careful review and comments concerning our manuscript entitled “Effect of heat treatment on microstructural evolution and microhardness change of Al-5Zn-0.03In-1Er alloy”. We corrected the mistakes in the manuscript according to your review. This manuscript has been checked and modified carefully and the revised portion are marked in red in the paper.

All in all, thank you for your careful review and comments concerning our manuscript entitled “Effect of heat treatment on microstructural evolution and microhardness change of Al-5Zn-0.03In-1Er alloy”. Best wishes to you.

Reviewer 5 Report

The work has been significantly improved, it can be published.

Author Response

(The authors gave the same response as above.)

Round 3

Reviewer 1 Report

The paper was improved. But the quality of the SEM images still needed to change. Fig 12 is still not suitable for the present investigation. Fig 13 needed more attention.

This manuscript is a resubmission of an earlier submission. The following is a list of the peer review reports and author responses from that submission.

Round 1

Reviewer 1 Report

1) The title needs to be corrected, because there is no need to indicate a specific research method in the title, it is better to pay attention to this in the methodology.

2) The abstract also needs to be corrected, it should be devoted to this article.

3) The choice of alloy and concentration of the additive also needs additional explanation. For what purposes can / is Al-Zn-In alloy applied? And what is the purpose of Er supplement research? Why is this concentration chosen?

4) "The effect of Er content in Al-Zn-In 41 alloys was studied and it was found that Er also plays a role in refining grains in Al-Zn-42 In alloys, but with the addition amount rising," Er is excellent grain modifier, not only in this group of alloys

5) "2) to study the evolution of Al3Er phase and how it affects 77 the Vickers hardness of the alloy." have not such studies been conducted before?

6) Figure 9 seems very strange. Please provide a detailed methodology for XRD analysis.

7) "X-ray diffraction (XRD) patterns of the UHTT alloys at 615 ℃ for various time are 205 shown in Figure 9. Compared with XRD patterns of the as-cast alloy, there is no obvious 206 difference in the peaks of α (Al) in alloys after UHTT. After the alloy was treated at 615 ℃ 207 for 16 h, the supersaturated solid solution decomposed, and more Al3Er phases precipi- 208 tated in the alloy. When the UHTT time exceeded 24 h, the content of Al3Er phases in the 209 alloy began to decrease as the UHTT temperature increased, and the precipitated phase 210 dissolved back into the α (Al) matrix. After UHTT at (615 ℃, 32 h), the Al3Er phases were 211 almost completely dissolved back into the α (Al) matrix. " Please explain this effect if the alloy is in a two-phase region at a given concentration and temperature

9) What is the confidence interval for the points in Figure 11?

10) It is necessary to add indices in the insert of Figure 12

11) The statement of tasks and the conclusion require adjustment and additional explanations.

"When UHTT is 615 ℃ × 32h, the Vickers hardness of the alloy increases by 335 15.52% than the as-cast alloy." what explains this change?

The main idea and result of this article, in my opinion, is that for better homogenization of the alloy, it must be heated as close as possible to the solidus temperature.

Author Response

Response to Reviewer 1 Comments

Dear Reviewer:

Thank you for your careful review and comments concerning our manuscript entitled “Effects of heat treatment on microstructural evolution and microhardness change of Al-5Zn-0.03In-1Er alloy”. Those comments are all valuable and very helpful for revising and improving our paper, as well as the important guiding significance to our researches. We have studied comments carefully and have made correction which we hope meet with approval. Revised portion are marked in red in the paper. The responds to your comments are as follows:

Point 1: The title needs to be corrected, because there is no need to indicate a specific research method in the title, it is better to pay attention to this in the methodology.

Response: Thank you for your suggestions. We have changed the title to” Effects of heat treatment on microstructural evolution and microhardness of Al-5Zn-0.03In-1Er alloy”. This title has been modified to be more accurate. And the methodology has been supplemented in line 105~111 of section 2.1. The additions are as follows:

“And the ultra-high temperature treatment (UHTT) temperature of Al-Zn-In-Er alloy was determined by differential scanning calorimetry (DSC). We chose the temperature that close to the solidus temperature as the heat treatment temperature. In order to explore the evolution of microstructure, heat treatments were carried out at different temperatures and different treatment time. Then the evolution of microstructure was investigated by OM, XRD, SEM, EDS and TEM.”

Point 2: The abstract also needs to be corrected, it should be devoted to this article.

Response: The abstract has been modified to be devoted to the paper. And We added accurate conclusion data in line20~28 of the abstract. The contents are as follows:

“The results showed that the main element Er is largely enriched in grain boundaries, but the changes of Zn, In are not obvious. When the temperature of UHTT is 614.3 ℃, the precipitated phase gradually dissolves into the matrix, and there are dispersed Al3Er particles in the crystal. And when UHTT is 615 ℃×32h, microhardness of the alloy is 32.5 HV, increased by 15.52% than that of as-cast.”

Point 3: The choice of alloy and concentration of the additive also needs additional explanation. For what purposes can / is Al-Zn-In alloy applied? And what is the purpose of Er supplement research? Why is this concentration chosen?

Response: According to our previous research [1-2], Al-5Zn-0.03In-xEr alloys have good electrochemical properties, and it is found that adding a small amount of Er element can refine the grains and improve the current efficiency of Al-Zn-In alloy, but when the Er content of the alloy exceeds 1wt. %, the self-corrosion of the alloy is intensified and the current efficiency is reduced. So, we chose Al-5Zn-0.03In-1Er as experimental alloy. The content we have added is shown in line 89~92:

“According to our previous research [7,8], Al-5Zn-0.03In-xEr alloys have good electrochemical properties, and it is found that when the content of Er is 1wt.%, the current efficiency of Al-Zn-In-Er alloy is the best. So, we chose Al-5Zn-0.03In-1Er as the experimental alloy.”

 Point 4: "The effect of Er content in Al-Zn-In alloys was studied and it was found that Er also plays a role in refining grains in Al-Zn- In alloys, but with the addition amount rising," Er is excellent grain modifier, not only in this group of alloys

Response: Absolutely yes. Z Jing [3] found that the addition of Er in Mg–9Zn–0.6Zr alloy improved deformability, as well as a fine and uniform microstructure. Xiao Wu [4] found that as the alloying element in die-cast ADC12 aluminum alloy Er can modify the eutectic silicon from a coarse plate-like and acicular structure to a fine branched and fibrous one. We can know that Er is excellent grain modifier from these articles. So, we can add Er in Al-Zn-In alloys and treat the alloy at ultra-high temperatures to explore the microstructure evolution to achieve the purpose of improving the properties of the alloy such as microhardness.

Point 5: "2) to study the evolution of Al3Er phase and how it affects 77 the Vickers hardness of the alloy." have not such studies been conducted before?

Response: The evolution of Al3Er phase has been investigated by Zhu S [5] in “Evolution of Al3Er phrase in Al-Er alloy”. The evolution of Al3Er phase in ultra-high temperature has not been studied yet, so we explored Al3Er evolution of the phase at ultra-high temperatures and effects of Al3Er on microhardness.

Point 6: Figure 9 seems very strange. Please provide a detailed methodology for XRD analysis.

Response: Figure 9. is X-ray diffraction (XRD) patterns of the UHTT alloys at 615 ℃ for various time. The peaks of α(Al) and Al3Er have been checked by PDF cards in Jade. Combined with Figure 8, the XRD peak intensity of Al3Er phase became weaker with the increase of treatment time. After UHTT at 615â„ƒ× 32h,the peaks of Al3Er phases nearly disappeared, but reprecipitated when the treatment time was increased to 40h. We can speculate the evolution of Al3Er phase with the treatment time increasing according to the microstructure evolution (Figure8) and changes of XRD peak intensity of Al3Er peaks in XRD patterns. The content We have modified is shown in line 245~255:

“X-ray diffraction (XRD) patterns of the UHTT alloys at 615 ℃ for various time are shown in Figure 8. Compared with XRD patterns of the as-cast alloy, there is no obvious difference in the peaks of α(Al) in alloys after UHTT. After the alloy was treated at 615 ℃ for 16 h, the peaks of Al3Er remained, the supersaturated solid solution decomposed, and more Al3Er phases precipitated in the alloy. When the UHTT time exceeded 24 h, the content of Al3Er phases in the alloy began to decrease as the UHTT time increased, and the precipitated phase dissolved back into the α(Al) matrix. After UHTT at (615 ℃, 32 h), the peaks of Al3Er phases nearly disappeared, the Al3Er phases were almost completely dissolved back into the α(Al) matrix. When the UHTT time was 40 h, the peaks of Al3Er reappeared, Al3Er phases reprecipitated from the matrix because solutes in matrix were depleted which reduced the solid solubility of A3Er phases.”

Point 7: "X-ray diffraction (XRD) patterns of the UHTT alloys at 615 ℃ for various time are 205 shown in Figure 9. Compared with XRD patterns of the as-cast alloy, there is no obvious 206 difference in the peaks of α (Al) in alloys after UHTT. After the alloy was treated at 615 ℃ 207 for 16 h, the supersaturated solid solution decomposed, and more Al3Er phases precipitated in the alloy. When the UHTT time exceeded 24 h, the content of Al3Er phases in the 209 alloy began to decrease as the UHTT temperature increased, and the precipitated phase 210 dissolved back into the α (Al) matrix. After UHTT at (615 ℃, 32 h), the Al3Er phases were 211 almost completely dissolved back into the α (Al) matrix. " Please explain this effect if the alloy is in a two-phase region at a given concentration and temperature

Response: According to the Al-Er binary phase diagram, when the Er content is nearly 1wt. %, Eutectic of both Al and Al3Er phases exists in the alloy. The EDS analysis in Figure 3. shows that Er element is enriched at the grain boundary. At last, we confirmed that the Er-rich phase is Al3Er and the matrix is α(Al)by peaks in XRD patterns.

Point 9: What is the confidence interval for the points in Figure 11?

Response: We have added error bars to the line chart in Figure 10. with microhardness standard deviation of the Al-5Zn-0.03In-1Er alloy under different heat treatment times.

Point 10: It is necessary to add indices in the insert of Figure 12

Response: For the accuracy of the results, The raw SAED patterns of the alloy after treatment at 615 ℃ for 32 h of Figure 12. has been indexed. According to the results of indices, the small particles dispersed in the matrix are Al3Er phase.

Point 11: The statement of tasks and the conclusion require adjustment and additional explanations.

Response: Adjustments of tasks and conclusion have been made in line 11-14 of abstract and line 394~396 of conclusion. The contents are as follows:

Abstract: “Adding an appropriate amount of Er element to the Al-Zn-In alloy can improve electrochemical performance of the Al alloy, but the Er segregation in the solid solution which reduces the com-prehensive properties of the alloy is difficult to eliminate, we found that the ultra-high temperature treatment (UHTT) can obviously reduce the segregation”

“Conclusion: In this study, the microstructure evolution and the microhardness change of Al-Zn-In-Er alloys was investigated after UHTT. Explanations for these phenomena are given. The conclusions could be drawn as follows:”

Point 12: "When UHTT is 615 ℃ × 32h, the Vickers hardness of the alloy increases by 15.52% than the as-cast alloy." what explains this change?

Response: It is speculated that with the increase of heat treatment time, Al3Er phase dissolves in the matrix, forming solid solution phenomenon that strengthening the microhardness. But when the time increases to 40h, Al3Er reprecipitates out of the matrix, and the effect of solid solution strengthening becomes weak resulting in lower microhardness. We have supplemented the cause of the change in line406~407 of conclusions. The contents are as follows:

“When UHTT is 615 ℃×32h, the microhardness of the alloy is 32.5 HV, and increases by 15.52% than that of as-cast alloy. The reason may be the solid solution of the Al3Er phase in Al-5Zn-0.03In-1Er alloy.”

Point 13: The main idea and result of this article, in my opinion, is that for better homogenization of the alloy, it must be heated as close as possible to the solidus temperature.

Response: Yes, this article is for better homogenization of the alloy. But due to the very slow diffusion rate of Er, we can only reduce the Er-rich segregation phase in the alloy with high temperature treatment as much as possible.

All in all, thank you for your review.

References

[1] Li H,Wei B,Xu ZB,Zeng J M,Chen R,Li H,Lu Y Y. Effect of Er on microstructure and electrochemical performance of Al-Zn-In anode [J]. Rare Metal Materials and Engineering ,2016,45(7):1848.

[2] Shen Z Chen X, Hang L I, et al. Effect of Er Content on the Microstructure and Current Efficiency of Al-Zn-In-xEr Alloy[J]. Foundry Technology.

[3] Jing Z, Ma R, Pan R. Effects of trace Er addition on the microstructure and mechanical properties of Mg–Zn–Zr alloy[J]. Materials & Design, 2010, 31(9):4043-4049.

[4] Xiao Wu, Hu, Fu gang, et al. Effects of rare earth Er additions on microstructure development and mechanical properties of die-cast ADC12 aluminum alloy[J]. Journal of Alloys and Compounds, 2012.

[5] Zhu S, Huang H, Nie Z, et al. Formation and evolution of Al3Er phrase in Al-Er alloy[J]. Chinese Journal of Rare Metals, 2009, 33(2):164-169.

Reviewer 2 Report

The authors investigated the homogenization annealing of the Al-Zn-In-Er alloy. Studies of the microstructure and hardness were carried out depending on the temperature and time of annealing. The article should be significantly improved, otherwise it cannot be published in the journal Metals.

    1. The title of the article must be corrected; the indication of research methods does not imply anything new.
    2. In the abstract, the authors describe the introductory part, it is better to describe the results.
    3. In fig. 1, the authors do not cite the presence of a phase with In, it is not clear where it is present in the composition.
    4. Line 118: “It is a typical as-cast microstructure, exhibiting serious dendritic segregation, which requires further heat treatment process to eliminate it.” - It is well known what is eliminated by homogenization.
    5. Figure 4 can be combined with Fig. 3 to save space.
    6. How is the alloy composition chosen? This requires an explanation.
    7. The purpose of the work is not clear. It should be noted the novelty of the work.
    8. It is necessary to explain the appearance and disappearance of the peaks in Fig. 7. What is this connected with? What is the reason for the different intensity of the peaks. Why is there no 625C, which was planned to be studied?
    9. What is the hardness measurement error and how significant are the results? Should be discussed.
    10. Is it possible to explain why the authors give the values of the activation energy from the literature, while there are no own results?
    11. Fig 12 must be indexed. Why are there so many dislocations after such a long annealing?

Author Response

Response to Reviewer 2 Comments

Dear Reviewer:

Thank you for your careful review and comments concerning our manuscript entitled “Effects of heat treatment on microstructural evolution and microhardness change of Al-5Zn-0.03In-1Er alloy”. Those comments are all valuable and very helpful for revising and improving our paper, as well as the important guiding significance to our researches. We have studied comments carefully and have made correction which we hope meet with approval. Revised portion are marked in red in the paper. The responds to your comments are as follows:

Point 1: The title of the article must be corrected; the indication of research methods does not imply anything new.

Response: Thank you for your suggestions. We have changed the title to” Effects of heat treatment on microstructural evolution and microhardness of Al-5Zn-0.03In-1Er alloy”. This title has been modified to be more accurate.

Point 2: In the abstract, the authors describe the introductory part, it is better to describe the results.

Response: The abstract has been modified. We added more detailed results in line20~28 of the abstract. The contents are as follows:

“The results showed that the main element Er is largely enriched in grain boundaries, but the changes of Zn, In are not obvious. When the temperature of UHTT is 614.3 ℃, the precipitated phase gradually dissolves into the matrix, and there are dispersed Al3Er particles in the crystal. And when UHTT is 615 ℃×32h, the microhardness of the alloy is 32.5 HV, increased by 15.52% than the as-cast alloy.”

Point 3:  In fig. 1, the authors do not cite the presence of a phase with In, it is not clear where it is present in the composition.

Response: It may be due to exceeding the detection accuracy of energy spectrum and XRD that In can’t be cited. We have made a supplementary explanation in line133~137 of section 3.1. The additions are as follows:

“The actual composition of the alloy in Table 1. tested by ICP-AES showed that there was In element in the alloy. The peaks of phase containing In element isn’t shown in Figure 1. Because its content is so low and uniformly distributed in the alloy that XRD couldn’t detect In, and the segregation of In can be neglected.”

Point 4: Line 118: “It is a typical as-cast microstructure, exhibiting serious dendritic segregation, which requires further heat treatment process to eliminate it.” - It is well known what is eliminated by homogenization.

Response: Yes, the purpose of the ultra-temperature treatment we used is to homogenize the alloy microstructure. But due to the very slow diffusion rate of Er, we can only reduce the Er-rich segregation phase in the alloy with high temperature treatment as much as possible.

Point 5: Figure 4 can be combined with Fig. 3 to save space.

Response: We have combined Fig 3. with Fig 4. and adjusted the serial numbers of all the diagrams behind Figure 3. Thank you for your suggestion.

Point 6: How is the alloy composition chosen? This requires an explanation

Response: According to our previous research [1-2], Al-5Zn-0.03In-xEr alloys have good electrochemical properties, and it is found that adding a small amount of Er element can refine the grains and improve the current efficiency of Al-Zn-In alloy, but when the Er content of the alloy exceeds 1wt. %, The self-corrosion of the alloy is intensified and the current efficiency is reduced. So, we chose Al-5Zn-0.03In-1Er as experimental alloy. The content We have added is shown in line89~92:

“According to our previous research [7,8], Al-5Zn-0.03In-xEr alloys have good electrochemical properties, and it is found that when the content of Er is 1wt.%, the current efficiency of Al-Zn-In-Er alloy is the best. So, we chose Al-5Zn-0.03In-1Er as the experimental alloy.”

Point 7: The purpose of the work is not clear. It should be noted the novelty of the work.

Response: Our purpose was mainly to reduce the Er-rich segregation phase in the alloy with high temperature treatment as much as possible. We added the purpose in line 12~14 of the abstract and in line 110~112 of section 2.1, The contents are as follows:

“But the Er segregation in the solid solution which reduces the comprehensive properties of the alloy is difficult to eliminate, we found that the ultra-high temperature treatment (UHTT) can obviously reduce the segregation”

“We chose high temperature that closes to the solidus temperature as the heat treatment temperature for reducing the segregation as much as possible.”

Point 8: It is necessary to explain the appearance and disappearance of the peaks in Fig. 7. What is this connected with? What is the reason for the different intensity of the peaks? Why is there no 625C, which was planned to be studied?

Response: We have explained the change of peaks of Al3Er phases and the reason that neglect the curve of 625 ℃ in line 210~224 of section 3.1. These peaks are connected with the evolution of Al3Er phases. And the intensity of different phases has a direct relationship with their content in the alloy. The contents added are as follows:

Why is no 625℃? “It can be seen from Figure 6. that a kind of spherical cage-like structure precipitates in the grain at 625 ℃, which accords with the characteristic of "overheating". In order to prevent the degradation of alloy properties caused by overheating, we chose 615 ℃ as the maximum heat treatment temperature.”

“When the UHTT temperature exceeded 595 ℃, the XRD peak intensity of Al3Er became weak because the content of Al3Er phases in the alloy began to decrease as the UHTT temperature increased, and precipitated phase dissolved back into the α(Al) matrix. After UHTT at (615 ℃, 24 h), combined with Figure 6e. and the peaks of Al3Er phases at 615 ℃, the Al3Er phases were almost completely dissolved back into the α(Al) matrix. “

Point 9: What is the hardness measurement error and how significant are the results? Should be discussed.

Response: We have added error bars to the line chart in Figure 10. with microhardness standard deviation of the Al-5Zn-0.03In-1Er alloy under different heat treatment times. The results shows Er addition can increase the microhardness of Al-5Zn-0.03In-1Er alloy. The reason may be the solid solution of the Al3Er phase.

Point 10: Is it possible to explain why the authors give the values of the activation energy from the literature, while there are no own results?

Response: In the section 3.5, We have adjusted the places of formulas of diffusion and relevant content to facilitate the combination of the activation energy obtained from literature and the formula. Fick's first law is used to prove the necessity of ultra-high temperature treatment for the diffusion of Er element. The comparison of different elements’ activation energy plays a role in proving the necessity of high temperature treatment for Er diffusion.

Point 11: Fig 12 must be indexed. Why are there so many dislocations after such a long annealing?

Response: For the accuracy of the results, The raw SAED patterns of the alloy after treatment at 615 ℃ for 32 h of Figure 12. has been indexed. According to the results of indices, the small particles dispersed in the matrix are Al3Er phase.

 It can be inferred from figure 11. that Al3Er solid solution into the grain boundary act as a strong obstacle to the dislocations motion, and fine Al3Er particles also hinder the movement of dislocation, both of which lead to the increase of microhardness and the existence of lots of dislocations. The more contents are as follows:

“The Microhardness of alloys is mainly dependent on three factors: grain size, strengthening effect, and second phases. The secondary phases gradually dissolved into the matrix, which resulted in a uniform distribution of residual granular secondary phases. These residual granular secondary phases could also act as a strong obstacle to the dislocation motion. Meanwhile, significant grain growth was not observed. Therefore, the strengthening effect was believed to be a dominant factor for the Microhardness at the initial stage of treatment.

Addition of alloying elements causes lattice distortion, and the precipitation phase has a significant anchoring effect (Figure 11.), both can hinder the movement of dislocations, which in turn produces a strengthening effect. So, the Microhardness increases at first, significantly.”

References

[1] Li H,Wei B,Xu ZB,Zeng J M,Chen R,Li H,Lu Y Y. Effect of Er on microstructure and electrochemical performance of Al-Zn-In anode [J]. Rare Metal Materials and Engineering ,2016,45(7):1848.

[2] Shen Z Chen X, Hang L I, et al. Effect of Er Content on the Microstructure and Current Efficiency of Al-Zn-In-xEr Alloy[J]. Foundry Technology.

Reviewer 3 Report

The paper is focused on the microstructural investigation of Al-Zn-In-Er alloys. The topic falls within the scope of the journal. The data are properly presented and discussed. On this basis, I recommend the publication after the following minor revisions:

  • 2.3. The scale length within SEM/OM images is not clearly readable. Please check and revise.
  • I suggest to determine the enthalpy changes for the melting by the analysis of the corresponding peak.
  • I recommend to present also the DSC curve in the heating run.

Author Response

Response to Reviewer 3 Comments

Dear Reviewer:

Thank you for your careful review and comments concerning our manuscript entitled “Effects of heat treatment on microstructural evolution and microhardness change of Al-5Zn-0.03In-1Er alloy”. Those comments are all valuable and very helpful for revising and improving our paper, as well as the important guiding significance to our researches. We have studied comments carefully and have made correction which we hope meet with approval. Revised portion are marked in red in the paper. The responds to your comments are as follows:

Point 1: 2.3. The scale length within SEM/OM images is not clearly readable. Please check and revise.

Response: Thank you for your careful check. I have carefully modified the scale length within SEM/OM images of Figure 2. and Figure 3.

Point 2: I suggest to determine the enthalpy changes for the melting by the analysis of the corresponding peak.

Response: Many studies on heat treatment for homogenization have use DSC curve to explore the evolution of the phases. For example, ChunMei Li[1] used DSC to find the melting points of α-η(MgZn2) and α-S(Al2CuMg) to explore the evolution of the phases. Wen Kai a[2] used DSC to determine the melting temperature of Mg(Zn, Cu, Al)2 phase to investigate the microstructural evolution of a high Zn-containing Al-Zn-Mg-Cu alloy during homogenization. So it’s necessary for us to determine the melting points of specific phase and solidus temperature of the alloy and the enthalpy changes is not necessary. Thank you for your suggestions.

Point 3: I recommend to present also the DSC curve in the heating run.

Response: I have added the DSC heating curve to Figure 5. Thank you for your review.

References

[1] ChunMei, Li, ZhiQian, et al. Intermetallic phase formation and evolution during homogenization and solution in Al-Zn-Mg-Cu alloys[J]. Science China Technological Sciences, 2013.

[2] A W K ,  A X B ,  A Z Y , et al. Microstructure Evolution of a High Zinc Containing Al-Zn-Mg-Cu Alloy during Homogenization[J]. Rare Metal Materials and Engineering, 2017, 46( 4):928-934.

Reviewer 4 Report

Accept in present form

Author Response

Thank you for your careful review!

Round 2

Reviewer 1 Report

1) according to Figure 4, the solidus temperature does not exceed 620 degrees. Please explain why such detailed work and detailed studies were carried out during homogenization at a temperature of 625?
2) why did the authors measure micro-hardness instead of a regular hardness test?
3) no response was received to point 6. Please provide detailed XRD analysis methodology for your samples.
4) no reply was received to point 7. Please explain the effect of dissolution and precipitation of the Al3ER phase if the alloy is in the two-phase region at a given concentration and temperature.
5) in Figure 8 there is a number of peaks not decoded, and this figure looks very strange.
6) in Fig. 9 it is very poorly visible from which particles the lines were obtained. Please make a similar image on larger images.
7) Figure 11 is not clear. What is the state of your material? Why is there so many dislocations in Figure 11? Give the size mark on the electron diffraction pattern.

Author Response

Response to Reviewer 1 Comments

Dear Reviewer:

Thank you for your careful review and comments concerning our manuscript entitled “Effects of heat treatment on microstructural evolution and microhardness change of Al-5Zn-0.03In-1Er alloy”. Those comments are all valuable and very helpful for revising and improving our paper, as well as the important guiding significance to our researches. We have studied comments carefully and have made correction which we hope meet with approval. Revised portion are marked in red in the paper. The responds to your comments are as follows:

Point 1: according to Figure 4, the solidus temperature does not exceed 620 degrees. Please explain why such detailed work and detailed studies were carried out during homogenization at a temperature of 625?

Response: Generally, the onset at 627.7 ℃ of the cooling DSC curve in Figure 4. is related to the initial melting point of the α(Al) matrix [1]. 625 ℃ should not exceed the solidus temperature.

Figure 5 f. shows that a kind of spherical cage-like structure precipitates in the grain at 625 ℃, which agrees with the feature of "overburnt structure". To prevent the degradation of alloy properties caused by overburning, we chose 615 ℃ as the maximum heat treatment temperature instead of 625℃. The evolution of the unknown structure and effects of the structure on the properties need to be explored to optimize the properties of Al alloys, so the heat treatment temperature at which the structure is formed is described.

Point 2: why did the authors measure micro-hardness instead of a regular hardness test?

Response: We used a Vickers microhardness tester and measured and calculated the diagonal of “diamond indentation” at low magnification to get microstructure values.

Generally, when exploring the evolution of microstructure of the alloy, the microhardness is usually tested to characterize the mechanical properties. And Vickers hardness test which used to measure microhardness has the highest accuracy among the commonly used hardness test methods. For example, Liu H [2] analyzed the microstructure evolution of Ti-6Al-4V alloy by SU5000 scanning electron microscope (SEM) and verified it by Vickers microhardness. The content we have added is shown in line 132~134:

“Figure 1. shows the selected areas. We selected areas under a microscope with a magnification of 200X and used the two diagonals of the diamond indentation to calculate the microhardness. The highest and lowest values in each group of data were omitted, and the average of the remaining three microhardness values was taken as the hardness value of the alloy”.

Point 3: no response was received to point 6. Please provide detailed XRD analysis methodology for your samples.

Response: We have calibrated the Miller index for XRD peaks in Figure 8. and made supplement to the XRD analysis of Figure 8. The content we have added is shown in line 261~271:

“Combined with the microstructure evolution process in Figure 8, the peaks in Figure 9. show that alloys after UHTT are mainly composed of α(Al) and Al3Er phases. Each characteristic peak is calibrated with the Miller index. Compared with the XRD pat-terns of the as-cast alloy, there is no obvious difference in the peaks of α(Al) in the al-loys after UHTT. After the alloy was treated at 615 ℃ for 16 h, the peaks of Al3Er re-mained and there is a distinct peak of Al3Er phase, the supersaturated solid solution decomposed, and more Al3Er phases precipitated in the alloy. When the UHTT time exceeded 24 h, the content of Al3Er phases in the alloy began to decrease as the UHTT time increased, and the precipitated phase dissolved back into the α(Al) matrix. After UHTT at (615 ℃, 32 h), the peaks of Al3Er phases nearly disappeared, the Al3Er phases were almost completely dissolved back into the α(Al) matrix or become extreme-ly small.”

 Point 4: no reply was received to point 7. Please explain the effect of dissolution and precipitation of the Al3Er phase if the alloy is in the two-phase region at a given concentration and temperature.

Response: The dissolution of Al3Er phase can make the microstructure of Al-5Zn-0.03In-1Er alloy more uniform, to reduce the potential difference between the second phase and the matrix and the number of tiny primary batteries produced in alloy microstructure. Therefore, the dissolution of Al3Er phase can effectively improve the electrochemical performance of the alloy. We also did some work to get the above result.

On the contrary, the precipitation of Al3Er phase increased the potential difference between the second phase and the matrix and the number of primary batteries produced in alloy microstructure. The electrochemical performance of the alloy was reduced.

Point 5: in Figure 8 there is a number of peaks not decoded, and this figure looks very strange.

Response: We have decoded each obvious peaks and calibrated the Miller index for XRD pattern peaks in figure 9.

Point 6: in Fig. 9 it is very poorly visible from which particles the lines were obtained. Please make a similar image on larger images.

Response: We found some errors by comparing figure 10. with the original data. Please give us some time to redo the line scan experiment.

Point 7: Figure 11 is not clear. What is the state of your material? Why is there so many dislocations in Figure 11? Give the size mark on the electron diffraction pattern.

Response: Figure 11. Shows a TEM image after 615℃ for 32h.

It can be inferred from Figure 12. that Al3Er solid solution into the grain boundary act as a strong obstacle to the dislocations motion, and fine Al3Er particles also hinder the movement of dislocation, both of which lead to the increase of microhardness and the existence of lots of dislocations. The more contents are as follows:

“The Microhardness of alloys is mainly dependent on three factors: grain size, strengthening effect, and second phases. The secondary phases gradually dissolved into the matrix, which resulted in a uniform distribution of residual granular secondary phases. These residual granular secondary phases could also act as a strong obstacle to the dislocation motion. Meanwhile, significant grain growth was not observed. Therefore, the strengthening effect was believed to be a dominant factor for the Microhardness at the initial stage of treatment.

Addition of alloying elements causes lattice distortion, and the precipitation phase has a significant anchoring effect (Figure 12.), both can hinder the movement of dislocations, which in turn produces a strengthening effect. So, the Microhardness increases at first, significantly.”

The size is marked on the electron diffraction pattern.

All in all, thank you for your review.

References

[1] Zhang, J.; Zuo, R.; Chen, Y.; Pan, F.; Luo, X. Microstructure evolution during homogenization of a τ-type Mg–Zn–Al alloy. Alloys Compd. 2008, 448, 316-320.

[2] Liu H, Zhang Z, K Xu, et al. Evolution of the α phase and microhardness for hot isostatic pressed Ti-6Al-4 V alloy during multi-pass deformation[J]. Materials Characterization, 2021, 178:111263.

[3] Shen Z , Chen X , Hang L I , et al. Effect of Er Content on the Microstructure and Current Efficiency of Al-Zn-In-xEr Alloy[J]. Foundry Technology.

Reviewer 2 Report

The authors made minor changes to the article, but this is not enough. The authors write that they have highlighted the corrections in red, but there are no such corrections in the text.
1. The title of the article remains too private, it is not clear why the title contains microhardness.

2. It is not clear why microhardness is included in the title.

3. The answer to Point 4 is insufficient. It is a known fact about homogenization and high temperature, the purpose of the study is not clear.

4. The answer to point 8 raises questions. Not sure if the judgment is correct.

5. From the change in hardness, it is noticeable that after 24 hours the hardness does not change, which contradicts the conclusions of the work.

6. Despite the answer, it is doubtful that such a content of dislocations after such a long annealing and a high temperature, what is the dislocation structure after a shorter and longer annealing time?

7. Please check Fig. 8, the disappearance of the most intense aluminum peak is impossible depending on the annealing time. There is a gross error in the figure.

Author Response

Response to Reviewer 2 Comments

Dear Reviewer:

Thank you for your careful review and comments concerning our manuscript entitled “Effects of heat treatment on microstructural evolution and microhardness change of Al-5Zn-0.03In-1Er alloy”. Those comments are all valuable and very helpful for revising and improving our paper, as well as the important guiding significance to our researches. We have studied comments carefully and have made correction which we hope meet with approval. Revised portion are marked in red in the paper. The responds to your comments are as follows:

Point 1: The title of the article remains too private; it is not clear why the title contains microhardness.

Response: We used a Vickers microhardness tester and measured and calculated the diagonal of “diamond indentation” at low magnification to get microstructure values.

Generally, when exploring the evolution of microstructure of the alloy, the microhardness is usually tested to characterize the mechanical properties. And Vickers hardness test which used to measure microhardness has the highest accuracy among the commonly used hardness test methods. For example, Liu H [2] analyzed the microstructure evolution of Ti-6Al-4V alloy by SU5000 scanning electron microscope (SEM) and verified it by Vickers microhardness. The content we have added is shown in line 132~134:

“Figure 1. shows the selected areas. We selected areas under a microscope with a magnification of 200X and used the two diagonals of the diamond indentation to calculate the microhardness. The highest and lowest values in each group of data were omitted, and the average of the remaining three microhardness values was taken as the hardness value of the alloy”.

Point 2: It is not clear why microhardness is included in the title.

Response: We used a Vickers microhardness tester and measured and calculated the diagonal of “diamond indentation” at low magnification to get microstructure values. Vickers hardness test has the highest accuracy among the commonly used hardness test methods. Therefore, we think that it is most appropriate to use the measured microhardness to represent the hardness of the alloy. If it is not appropriate, we will make modifications.

Point 3: The answer to Point 4 is insufficient. It is a known fact about homogenization and high temperature, the purpose of the study is not clear.

Response: Er usually exists in the form of Al3Er compound phase in Al-Zn-In alloy. The homogenization and high temperature can make Al3Er phase dissolve back into α(Al) matrix. The dissolution of Al3Er phase can effectively improve the electrochemical performance of the alloy. We also did some work to get the above result. The content of purpose we have added is shown in line 72~76 of Introduction:

“Therefore, to explore the evolution of the phases with increasing temperature and time, and its precipitation mechanism, UHTT was used in this work to achieve the purpose of homogenizing the alloy. These were basic works for the research group to study the electrochemical behavior of alloys in the future“.

Point 4: The answer to point 8 raises questions. Not sure if the judgment is correct.

Response: These peaks are connected with the evolution of Al3Er phases. XRD patterns combined with figure 6. show that the appearance and disappearance of Al3Er peak intense can determine the dissolution and precipitation of Al3Er phase.

The dissolution of Al3Er phase can make the microstructure of Al-5Zn-0.03In-1Er alloy more uniform, to reduce the potential difference between the second phase and the matrix and the number of tiny primary batteries produced in alloy microstructure. Therefore, the dissolution of Al3Er phase can effectively improve the electrochemical performance of the alloy. We also did some work to get the above result.

On the contrary, the precipitation of Al3Er phase increased the potential difference between the second phase and the matrix and the number of primary batteries produced in alloy microstructure. The electrochemical performance of the alloy was reduced.

Point 5: From the change in hardness, it is noticeable that after 24 hours the hardness does not change, which contradicts the conclusions of the work.

Response: We choose the proper heat treatment on the basis of microstructure evolution and XRD analysis instead of microhardness. Although the microhardness does not change much after 24 h heat treatment, Figure 9. shows Al3Er phase is less or smaller  at 32h treatment. The homogenization effect of the alloy after 615℃×32h is better.  

The hardness of alloys increased after long annealing time. In order to find the cause of the increase of hardness, the samples were analyzed by TEM. As a result, a large number of dislocations were found in alloy after treatment of 615℃×32h. It may be that nanoscale Al3Er formed from the dissolution of the second phase produces pinning effect on relative dislocations at grain boundaries to cause the dislocation tangle in Figure 12.

Point 6: Despite the answer, it is doubtful that such a content of dislocations after such a long annealing and a high temperature, what is the dislocation structure after a shorter and longer annealing time?

Response: We only made the best homogenized TEM samples of 615℃×32h.The main purpose of the TEM image is the identification of particle Al3Er phase and the causes of increase of hardness. If the TEM images of each heat treatment state are required, please give us around ten days to supplement the TEM analysis to conduct a comparative experiment.

Point 7: Please check Fig. 8, the disappearance of the most intense aluminium peak is impossible depending on the annealing time. There is a gross error in the figure.

Response: We calibrated the peaks that had not been calibrated before.After we compared XRD patterns with latest phase analysis PDF cards by “Jade” software, the peaks of around 82.6 °at 16 h and 44.93 °at 40 h are α(Al) peaks. The diffraction angles of α(Al) peaks in different alloys states are not necessarily equal.

All in all, thank you for your review.

References

[1] Liu H, Zhang Z, K Xu, et al. Evolution of the α phase and microhardness for hot isostatic pressed Ti-6Al-4 V alloy during multi-pass deformation[J]. Materials Characterization, 2021, 178:111263.

Round 3

Reviewer 1 Report

the authors made some changes, but the main questions were left without attention and the answer to them has not been received.
for example
1) why does the Al3Ep phase first dissolve and then precipitate again during annealing?
2) the questionable data in Figure 9 was never explained.
3) Figure 10 requires revision
etc

Reviewer 2 Report

The authors have made only minor changes in the article, but the article cannot be published as it has serious errors.  Authors should revise the results, especially XRD plots and analyze the datas.